# Zn²⁺ is essential for Ca²⁺ oscillations in mouse eggs

Hiroki Akizawa[1], Emily M Lopes[1,2], Rafael A Fissore[1]*

[1]Department of Veterinary and Animal Sciences, University of Massachusetts Amherst, Amherst, United States; [2]Molecular and Cellular Biology Graduate Program, University of Massachusetts, Amherst, United States

**Abstract** Changes in the intracellular concentration of free calcium ($Ca^{2+}$) underpin egg activation and initiation of development in animals and plants. In mammals, the $Ca^{2+}$ release is periodical, known as $Ca^{2+}$ oscillations, and mediated by the type 1 inositol 1,4,5-trisphosphate receptor ($IP_3R1$). Another divalent cation, zinc ($Zn^{2+}$), increases exponentially during oocyte maturation and is vital for meiotic transitions, arrests, and polyspermy prevention. It is unknown if these pivotal cations interplay during fertilization. Here, using mouse eggs, we showed that basal concentrations of labile $Zn^{2+}$ are indispensable for sperm-initiated $Ca^{2+}$ oscillations because $Zn^{2+}$-deficient conditions induced by cell-permeable chelators abrogated $Ca^{2+}$ responses evoked by fertilization and other physiological and pharmacological agonists. We also found that chemically or genetically generated eggs with lower levels of labile $Zn^{2+}$ displayed reduced $IP_3R1$ sensitivity and diminished ER $Ca^{2+}$ leak despite the stable content of the stores and $IP_3R1$ mass. Resupplying $Zn^{2+}$ restarted $Ca^{2+}$ oscillations, but excessive $Zn^{2+}$ prevented and terminated them, hindering $IP_3R1$ responsiveness. The findings suggest that a window of $Zn^{2+}$ concentrations is required for $Ca^{2+}$ responses and $IP_3R1$ function in eggs, ensuring optimal response to fertilization and egg activation.

*For correspondence:
rfissore@umass.edu

Competing interest: The authors declare that no competing interests exist.

## eLife assessment

This article reports an **important** series of results showing the relationship between oscillatory zinc and calcium fluctuations during egg activation and fertilization. **Compelling** evidence using several complimentary approaches provides further insight into the signals for proper egg activation that underpin successful fertilization and embryo development. The findings are significant because they may lead to improvements in assisted reproduction methods.

## Introduction

Vertebrate eggs are arrested at the metaphase stage of the second meiosis (MII) when ovulated because they have an active Cdk1/cyclin B complex and inactive APC/C^Cdc20 (*Heim et al., 2018*). Release from MII initiates egg activation, the first hallmark of embryonic development (*Ducibella et al., 2002*; *Schultz and Kopf, 1995*). The universal signal of egg activation is an increase in the intracellular concentration of calcium ($Ca^{2+}$) (*Ridgway et al., 1977*; *Stricker, 1999*). $Ca^{2+}$ release causes the inactivation of the APC/C inhibitor Emi2, which enhances cyclin B degradation and induces meiotic exit (*Lorca et al., 1993*; *Shoji et al., 2006*; *Suzuki et al., 2010a*). In mammals, the stereotypical fertilization $Ca^{2+}$ signal, oscillations, consists of transient but periodical $Ca^{2+}$ increases that promote progression into interphase (*Deguchi et al., 2000*; *Miyazaki et al., 1986*). The sperm-borne phospholipase C zeta1 (PLC ζ) persistently stimulates the production of inositol 1,4,5-trisphosphate ($IP_3$) (*Matsu-ura et al., 2019*; *Saunders et al., 2002*; *Wu et al., 2001*) that binds its cognate receptor in the endoplasmic reticulum (ER), $IP_3R1$, and causes $Ca^{2+}$ release from the egg's main $Ca^{2+}$ reservoir

(*Wakai et al., 2019*). The intake of extracellular $Ca^{2+}$ via plasma membrane channels and transporters ensures the persistence of the oscillations (*Miao et al., 2012*; *Stein et al., 2020*; *Wakai et al., 2019*; *Wakai et al., 2013*).

Before fertilization, maturing oocytes undergo cellular and biochemical modifications (see for review *Ajduk et al., 2008*). The nucleus of immature oocytes, known as the germinal vesicle (GV), undergoes the breakdown of its envelope, marking the onset of maturation and setting in motion a series of cellular events that culminate with the release of the first polar body, the correct ploidy for fertilization, and re-arrest at MII (*Eppig, 1996*). Other organelles are also reorganized, such as cortical granules migrate to the cortex for exocytosis and polyspermy block, mitochondria undergo repositioning, and the cytoplasm's redox state becomes progressively reduced to promote the exchange of the sperm's protamine load (*Liu, 2011*; *Perreault et al., 1988*; *Wakai et al., 2014*). Wide-ranging adaptations also occur in the $Ca^{2+}$ release machinery to produce timely and protracted $Ca^{2+}$ oscillations following sperm entry (*Fujiwara et al., 1993*; *Lawrence et al., 1998*), including the increase in the content of the $Ca^{2+}$ stores, ER reorganization with cortical cluster formation, and increased $IP_3R1$ sensitivity (*Lee et al., 2006*; *Wakai et al., 2012*). The total intracellular levels of zinc ($Zn^{2+}$) also remarkably increase during maturation, amounting to a 50% rise, which is necessary for oocytes to proceed to the telophase I of meiosis and beyond (*Kim et al., 2010*). Notably, after fertilization, $Zn^{2+}$ levels need to decrease, as Emi2 is a $Zn^{2+}$-associated molecule, and high $Zn^2$ levels prevent MII exit (*Bernhardt et al., 2012*; *Shoji et al., 2014*; *Suzuki et al., 2010b*). Following the initiation of $Ca^{2+}$ oscillations, approximately 10–20% of the $Zn^{2+}$ accrued during maturation is ejected during the $Zn^{2+}$ sparks, a conserved event in vertebrates and invertebrate species (*Converse and Thomas, 2020*; *Kim et al., 2011*; *Mendoza et al., 2022*; *Que et al., 2019*; *Seeler et al., 2021*; *Tokuhiro and Dean, 2018*; *Wozniak et al., 2020*; *Zhang et al., 2016*). The use of $Zn^{2+}$ chelators such as N,N,N,N-tetrakis (2-pyridinylmethyl)–1,2-ethylenediamine (TPEN) to create $Zn^{2+}$-deficient conditions buttressed the importance of $Zn^{2+}$ during meiotic transitions (*Kim et al., 2010*; *Suzuki et al., 2010b*). However, whether the analogous dynamics of $Ca^{2+}$ and $Zn^{2+}$ during maturation imply crosstalk and $Zn^{2+}$ levels modulate $Ca^{2+}$ release during fertilization is unknown.

$IP_3Rs$ are the most abundant intracellular $Ca^{2+}$ release channel in non-muscle cells (*Berridge, 2016*). They form a channel by assembling into tetramers with each subunit of ~270 kDa MW (*Taylor and Tovey, 2010*). Mammalian eggs express the type I $IP_3R$, the most widespread isoform (*Fissore et al., 1999*; *Parrington et al., 1998*). $IP_3R1$ is essential for egg activation because its inhibition precludes $Ca^{2+}$ oscillations (*Miyazaki and Ito, 2006*; *Miyazaki et al., 1992*; *Xu et al., 2003*). Myriad and occasionally cell-specific factors influence $Ca^{2+}$ release through the $IP_3R1$ (*Taylor and Tovey, 2010*). For example, following fertilization, $IP_3R1$ undergoes ligand-induced degradation caused by the sperm-initiated long-lasting production of $IP_3$ that effectively reduces the $IP_3R1$ mass (*Brind et al., 2000*; *Jellerette et al., 2000*). Another regulatory mechanism is $Ca^{2+}$, a universal cofactor, which biphasically regulates $IP_3Rs$' channel opening (*Iino, 1990*; *Jean and Klee, 1986*), congruent with several $Ca^{2+}$ and calmodulin binding sites on the channel's sequence (*Sienaert et al., 1997*; *Sipma et al., 1999*). Notably, $Zn^{2+}$ may also participate in $IP_3R1$ regulation. Recent studies using electron cryomicroscopy (cryoEM), a technique that allows peering into the structure of $IP_3R1$ with a near-atomic resolution, have revealed that a helical linker (LNK) domain near the C-terminus mediates the coupling between the N- and C-terminal ends necessary for channel opening (*Fan et al., 2015*). The LNK domain contains a putative zinc-finger motif proposed to be vital for $IP_3R1$ function (*Fan et al., 2015*; *Paknejad and Hite, 2018*). Therefore, the exponential increase in $Zn^{2+}$ levels in maturing oocytes, besides its essential role in meiosis progression, may optimize the $IP_3R1$ function, revealing hitherto unknown cooperation between these cations during fertilization.

Here, we examined whether crosstalk between $Ca^{2+}$ and $Zn^{2+}$ is required to initiate and sustain $Ca^{2+}$ oscillations and maintain $Ca^{2+}$ store content in MII eggs. We found that $Zn^{2+}$-deficient conditions inhibited $Ca^{2+}$ release and oscillations without reducing $Ca^{2+}$ stores, $IP_3$ production, $IP_3R1$ expression, or altering the viability of eggs or zygotes. We show instead that $Zn^{2+}$ deficiency impaired $IP_3R1$ function and lessened the receptor's ability to gate $Ca^{2+}$ release out of the ER. Remarkably, resupplying $Zn^{2+}$ re-established the oscillations interrupted by low $Zn^{2+}$, although persistent increases in intracellular $Zn^{2+}$ were harmful, disrupting the $Ca^{2+}$ responses and preventing egg activation. Together, the results show that besides contributing to oocyte maturation, $Zn^{2+}$ has a central function in $Ca^{2+}$ homeostasis

such that optimal $Zn^{2+}$ concentrations ensure $IP_3R1$ function and the $Ca^{2+}$ oscillations required for initiating embryo development.

## Results

### TPEN dose-dependently lowers intracellular $Zn^{2+}$ and inhibits sperm-initiated $Ca^{2+}$ oscillations

TPEN is a cell-permeable, non-specific chelator with a high affinity for transition metals widely used to study their function in cell physiology (*Arslan et al., 1985*; *Lo et al., 2020*). Mouse oocytes and eggs have exceedingly high intracellular concentrations of $Zn^{2+}$ (*Kim et al., 2011*; *Kim et al., 2010*), and the TPEN-induced defects in the progression of meiosis have been ascribed to its chelation (*Bernhardt et al., 2011*; *Kim et al., 2010*). In support of this view, the $Zn^{2+}$ levels of cells showed acute reduction after TPEN addition, as reported by indicators such as FluoZin-3 (*Arslan et al., 1985*; *Gee et al., 2002*; *Suzuki et al., 2010b*). Studies in mouse eggs also showed that the addition of low µM (40–100) concentrations of TPEN disrupted $Ca^{2+}$ oscillations initiated by fertilization or $SrCl_2$ (*Lawrence et al., 1998*; *Suzuki et al., 2010b*), but the mechanism(s) and target(s) of the inhibition remained unknown. To gain insight into this phenomenon, we first performed dose-titration studies to determine the effectiveness of TPEN in lowering $Zn^{2+}$ in eggs. The addition of 2.5 µM TPEN protractedly reduced $Zn^{2+}$ levels, whereas 5 and 10 µM TPEN acutely and persistently reduced FluoZin-3 fluorescence (*Figure 1A*). These concentrations of TPEN are higher than the reported free $Zn^{2+}$ concentrations in cells, but within range of those found in typical culture conditions (*Lo et al., 2020*; *Qin et al., 2011*). We next determined the concentrations of TPEN required to abrogate fertilization-initiated oscillations. Following intracytoplasmic sperm injection (ICSI), we monitored $Ca^{2+}$ responses while increasing TPEN concentrations. As shown in *Figure 1B*, 5 and 10 µM TPEN effectively blocked ICSI-induced $Ca^{2+}$ oscillations in over half of the treated cells, and the remaining eggs, after a prolonged interval, resumed lower-frequency rises (*Figure 1B*, center panels). Finally, 50 µM or greater concentrations of TPEN permanently blocked these oscillations (*Figure 1B*, right panel). It is noteworthy that at the time of addition TPEN concentrations of 5 µM or above induce a sharp drop in basal Fura-2 F340/F380 ratios, consistent with Fura-2's high affinity for $Zn^{2+}$ (*Snitsarev et al., 1996*).

We next used membrane-permeable and -impermeable chelators to assess whether TPEN inhibited $Ca^{2+}$ oscillations by chelating $Zn^{2+}$ from intracellular or extracellular compartments. The addition of the high-affinity but cell-impermeable $Zn^{2+}$ chelators DTPA and EDTA neither terminated nor temporarily interrupted ICSI-induced $Ca^{2+}$ oscillations (*Figure 1C*), although protractedly slowed them down, possibly because of chelation and lowering of external $Ca^{2+}$ (*Figure 1C*). These results suggest that chelation of external $Zn^{2+}$ does not affect the continuation of oscillations. We cannot determine that EDTA successfully chelated all external $Zn^{2+}$, but the evidence that the addition of EDTA to the monitoring media containing cell-impermeable FluoZin-3 caused a marked reduction in fluorescence suggests that a noticeable fraction of the available $Zn^{2+}$ was sequestered (*Figure 1—figure supplement 1*). Similarly, injection of m*Plcz1* mRNA in eggs incubated in $Ca^{2+}$ and $Mg^{2+}$-free media supplemented with EDTA, to maximize the chances of chelation of external $Zn^{2+}$, initiated low-frequency but persistent oscillations, and addition of $Ca^{2+}$ and $Mg^{2+}$ restored the physiological periodicity (*Figure 1—figure supplement 1*). Lastly, another $Zn^{2+}$-permeable chelator, TPA, blocked the ICSI-initiated $Ca^{2+}$ oscillations but required higher concentrations than TPEN (*Figure 1D*). Collectively, the data suggest that basal levels of labile internal $Zn^{2+}$ are essential to sustain the fertilization-initiated $Ca^{2+}$ oscillations in eggs.

We next evaluated whether $Zn^{2+}$ depletion prevented the completion of meiosis and pronuclear (PN) formation. To this end, ICSI-fertilized eggs were cultured in the presence of 10 µM TPEN for 8 hr, during which the events of egg activation were examined (*Figure 1E* and *Table 1*). All fertilized eggs promptly extruded second polar bodies regardless of treatment (*Figure 1E*). TPEN, however, impaired PN formation, and by 4 or 7 hr post-ICSI, most treated eggs failed to show PNs, unlike controls (*Figure 1E* and *Table 1*). Together, these results demonstrate that depletion of $Zn^{2+}$ terminates $Ca^{2+}$ oscillations and delays or prevents events of egg activation, including PN formation.

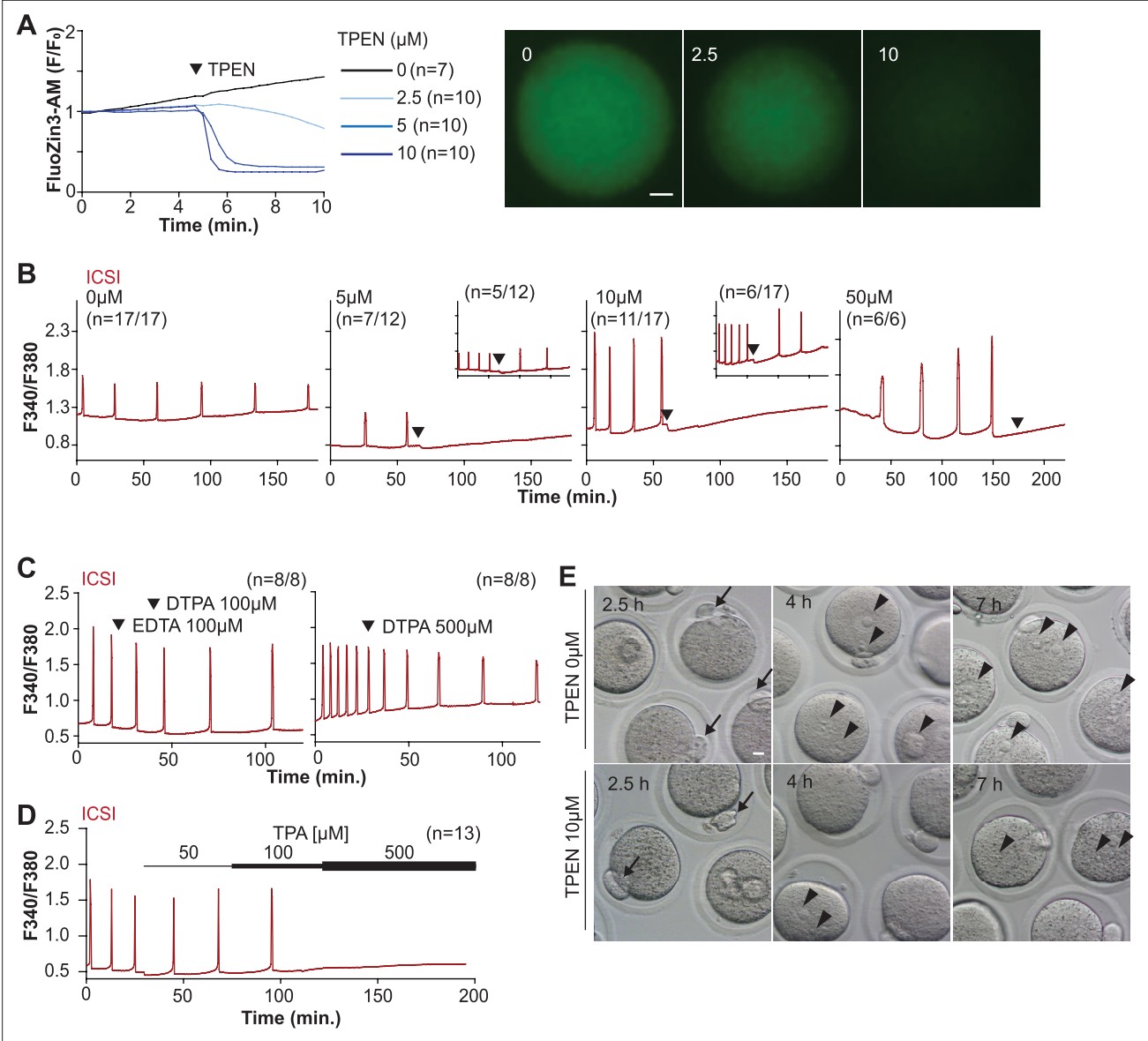

**Figure 1.** TPEN-induced Zn²⁺ deficiency inhibits fertilization-initiated Ca²⁺ oscillations in a dose-dependent manner. (**A**) Left panel: representative normalized Zn²⁺ recordings of MII eggs loaded with FluoZin-3AM following the addition of increasing concentrations of TPEN (0 µM, DMSO, black trace; 2.5 µM, sky blue; 5 µM, blue; 10 µM, navy). TPEN was directly added to the monitoring media. Right panel: representative fluorescent images of MII eggs loaded FluoZin-3AM supplemented with 0, 2.5, and 10 µM of TPEN. Scale bar: 10 µm. (**B–D**) (**B**) Representative Ca²⁺ oscillations following intracytoplasmic sperm injection (ICSI) after the addition of 0, 5, 10, or 50 µM TPEN (arrowheads). Insets show representative traces for eggs that resumed Ca²⁺ oscillations after TPEN. (**C**) As above, but following the addition of 100 µM EDTA, 100 or 500 µM DTPA (time of addition denoted by arrowheads). (**D**) Ca²⁺ oscillations following ICSI after the addition of 50, 100, and 500 µM TPA (horizontal bars of increasing thickness). (**E**) Representative bright field images of ICSI-fertilized eggs 2.5, 4, and 7 hr after sperm injection. Arrows and arrowheads denote the second polar body and pronuclear (PN) formation, respectively. Scale bar: 10 µm.

The online version of this article includes the following figure supplement(s) for figure 1:

**Figure supplement 1.** Cell-impermeable chelators effectively reduce Zn²⁺ levels in external media but do prevent initiation or continuation of Ca²⁺ oscillations.

## TPEN is a universal inhibitor of Ca²⁺ oscillations in eggs

Mammalian eggs initiate Ca²⁺ oscillations in response to numerous stimuli and conditions (*Miyazaki and Ito, 2006*; *Wakai and Fissore, 2013*). Fertilization and its release of PLC ζ stimulate the phosphoinositide pathway, producing IP₃ and Ca²⁺ oscillations (*Miyazaki, 1988*; *Saunders et al., 2002*). Neurotransmitters such as acetylcholine (Ach) and other G-protein-coupled receptor agonists engage

**Table 1.** Addition of TPEN after intracytoplasmic sperm injection (ICSI) does not prevent extrusion of the second polar body but precludes pronuclear (PN) formation.

| Group* | No. of zygotes | Second polar body (2.5 hr) | PN | |
| --- | --- | --- | --- | --- |
| | | | 4 hr | 7 hr |
| Untreated | 26 | 25 (96.1%) | 23 (88.5%) | 23 (88.5%) |
| TPEN (10 μM) | 27 | 24 (88.9%) | 1 (3.7%)*** | 2 (7.4%)*** |

***p<0.001.
*Data from three different replicates for each group.

a similar mechanism (*Dupont et al., 1996*; *Kang et al., 2003*), although in these cases, $IP_3$ production occurs at the plasma membrane and is short-lived (*Kang et al., 2003*; *Swann and Parrington, 1999*). Agonists such as $SrCl_2$ and thimerosal generate oscillations by sensitizing $IP_3R1$ without producing $IP_3$. The mechanism(s) of $SrCl_2$ is unclear, although its actions are reportedly directly on the $IP_3R1$ (*Hajnóczky and Thomas, 1997*; *Hamada et al., 2003*; *Nomikos et al., 2015*; *Nomikos et al., 2011*; *Sanders et al., 2018*). Thimerosal oxidizes dozens of thiol groups in the receptor, which enhances the receptor's sensitivity and ability to release $Ca^{2+}$ (*Bootman et al., 1992*; *Evellin et al., 2002*; *Joseph et al., 2018*). We took advantage of the varied points at which the mentioned agonists engage the phosphoinositide pathway to examine TPEN's effectiveness in inhibiting their effects. m*Plcz1* mRNA injection, like fertilization, induces persistent $Ca^{2+}$ oscillations, although m*Plcz1*'s tends to be more robust. Consistent with this, the addition of 10 and 25 μM TPEN transiently interrupted or belatedly terminated oscillations, whereas 50 μM acutely stopped all responses (*Figure 2A*). By contrast, $SrCl_2$-initiated rises were the most sensitive to $Zn^{2+}$-deficient conditions, with 2.5 μM TPEN nearly terminating all oscillations that 5 μM did (*Figure 2B*). TPEN was equally effective in ending the Ach-induced $Ca^{2+}$ responses (*Figure 2C*), but curbing thimerosal responses required higher concentrations (*Figure 2D*). Lastly, we ruled out that downregulation of $IP_3R1$ was responsible for the slow-down or termination of the oscillations by TPEN. To accomplish this, we examined the $IP_3R1$ mass in eggs (*Jellerette et al., 2004*) with and without TPEN supplementation and injection of m*Plcz1* mRNA. By 4 hr post-injection, *Plcz1* induced the expected downregulation of $IP_3R1$ reactivity, whereas it was insignificant in TPEN-treated and *Plcz1* mRNA-injected eggs, as it was in uninjected control eggs (*Figure 2F*). These findings together show that $Zn^{2+}$ deficiency inhibits the $IP_3R1$-mediated $Ca^{2+}$ oscillations independently of $IP_3$ production or loss of receptor, suggesting a role of $Zn^{2+}$ on $IP_3R1$ function (*Figure 2E*).

## $Zn^{2+}$ depletion reduces $IP_3R1$-mediated $Ca^{2+}$ release

To directly assess the inhibitory effects of TPEN on $IP_3R1$ function, we used caged $IP_3$ ($cIP_3$) that, after short UV pulses, releases $IP_3$ into the ooplasm (*Wakai et al., 2012*; *Walker et al., 1987*). To exclude the possible contribution of external $Ca^{2+}$ to the responses, we performed the experiments in $Ca^{2+}$-free media. In response to sequential $cIP_3$ release 5 min apart, control eggs displayed corresponding $Ca^{2+}$ rises that occasionally transitioned into short-lived oscillations (*Figure 3A*). The addition of TPEN after the third $cIP_3$ release prevented the subsequent $Ca^{2+}$ response and prematurely terminated the in-progress $Ca^{2+}$ rises (*Figure 3B*, inset). Pre-incubation of eggs with TPEN precluded $cIP_3$-induced $Ca^{2+}$ release, even after 5 s UV exposure (*Figure 3C*). The addition of excess $ZnSO_4$ (100 μM) overcame TPEN's inhibitory effects only if added before (*Figure 3E*) and not after the addition of TPEN (*Figure 3D*). Similar concentrations of $MgCl_2$ or $CaCl_2$ failed to reverse TPEN effects (*Figure 3F and G*). Together, the results show that $Zn^{2+}$ is required for $IP_3R1$-mediated $Ca^{2+}$ release downstream of $IP_3$ production, appearing to interfere with receptor gating, as suggested by TPEN's rapid termination of in-progress $Ca^{2+}$ rises and ongoing oscillations.

ERp44 is an ER luminal protein of the thioredoxin family that interacts with the $IP_3R1$, reportedly inhibiting its ability to mediate $Ca^{2+}$ release (*Higo et al., 2005*). The localization of ERp44 in the ER-Golgi intermediate compartment of somatic cells correlates with $Zn^{2+}$'s availability and changes dramatically after TPEN treatment (*Higo et al., 2005*; *Watanabe et al., 2019*). To rule out the possibility that TPEN suppresses the function of $IP_3R1$ by modifying the subcellular distribution of ERp44, we overexpressed ERp44 by injecting mRNA encoding HA-tagged ERp44 into MII eggs and monitored the effect on $Ca^{2+}$ release. TPEN did not alter the localization of ERp44 (*Figure 3—figure*

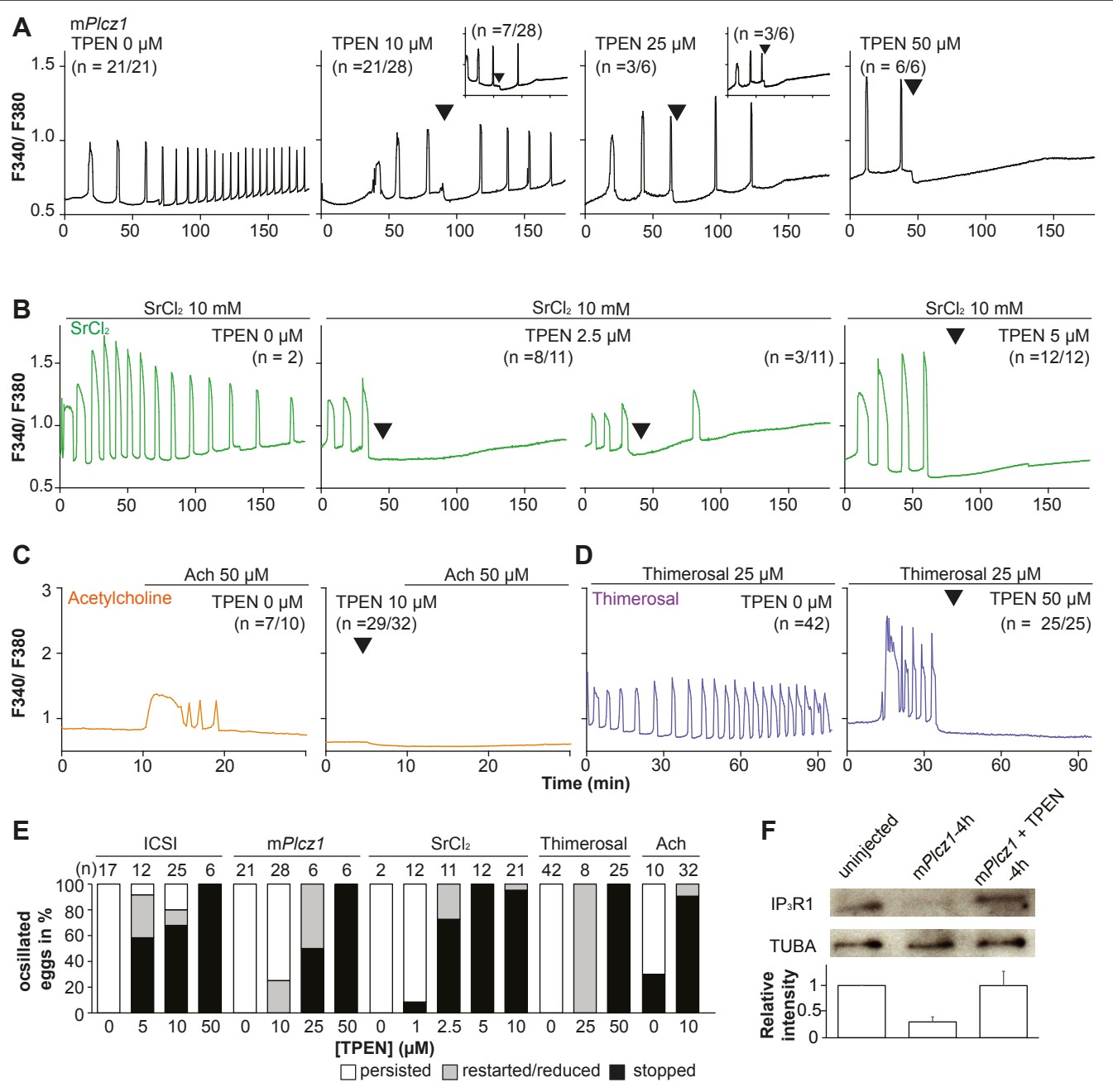

**Figure 2.** TPEN dose-dependently inhibits $Ca^{2+}$ oscillations in eggs triggered by a broad spectrum of agonists that stimulate the PI pathway or $IP_3R1$. (**A–D**) Representative $Ca^{2+}$ responses induced by (**A**) m*Plcz1* mRNA microinjection (0.01 μg/μl, black traces), (**B**) strontium chloride (10 mM, green), (**C**) acetylcholine chloride (50 μM, orange), and (**D**) thimerosal (25 μM, purple) in MII eggs. Increasing concentrations of TPEN were added to the monitoring media (arrowheads above traces denotes the time of adding). Insets in the upper row show representative traces of eggs that stop oscillating despite others continuing to oscillate. (**E**) Each bar graph summarizes the TPEN effect on $Ca^{2+}$ oscillations at the selected concentrations for each of the agonists in (**A–D**). (**F**) Western blot showing the intensities of $IP_3R1$ and alpha-tubulin bands in MII eggs or in eggs injected with m*Plcz1* mRNA and incubated or not with TPEN above (p<0.01). Thirty eggs per lane in all cases. This experiment was repeated twice, and the mean relative intensity of each blot is shown in the bar graph below.

The online version of this article includes the following source data for figure 2:

**Source data 1.** $IP_3R1$ and TUBA western blottings.

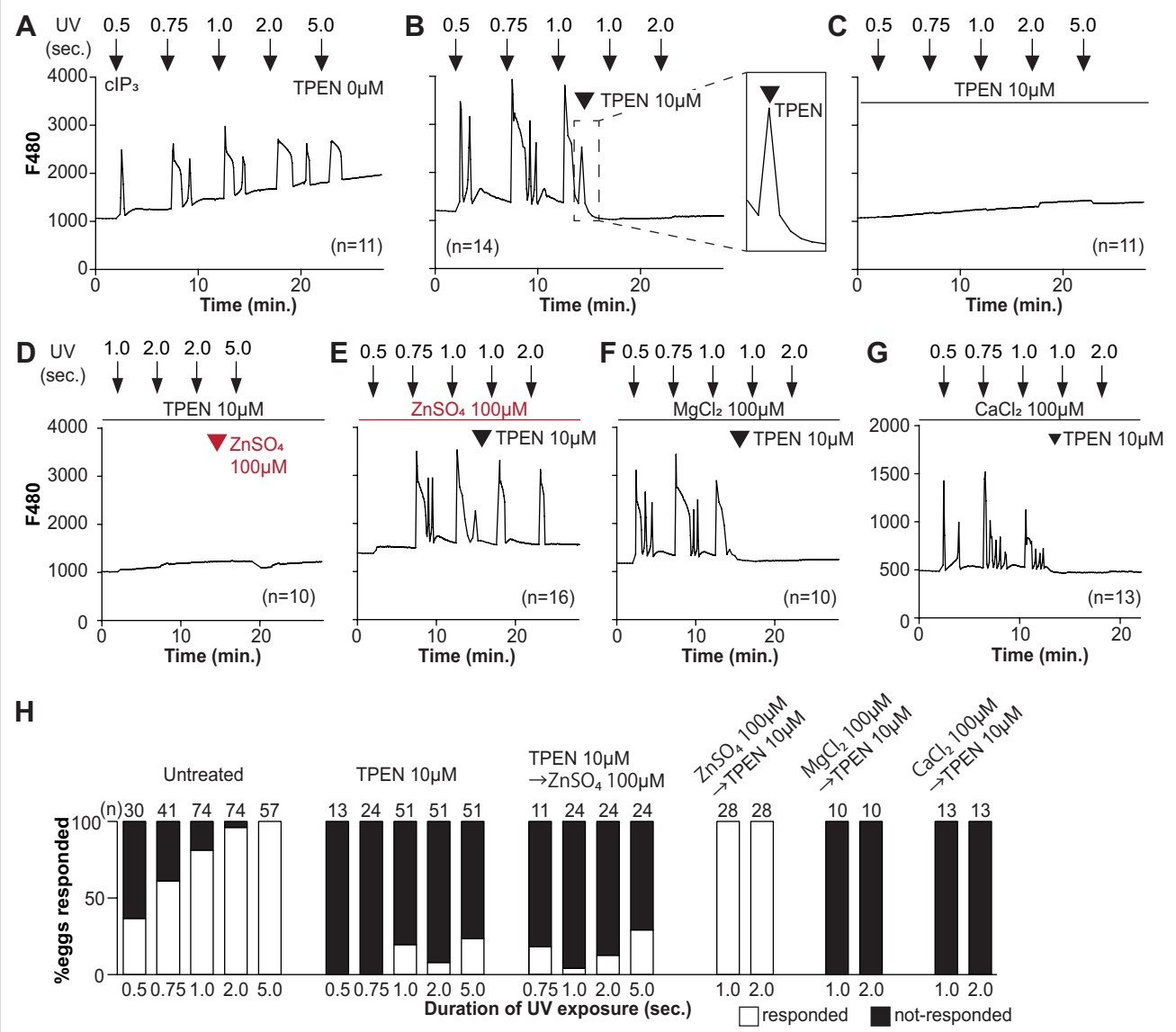

**Figure 3.** TPEN inhibition of cIP$_3$-induced Ca$^{2+}$ release is precluded by ZnSO$_4$ supplementation before TPEN exposure. (**A–G**) Representative Ca$^{2+}$ responses in MII eggs triggered by the release of caged IP$_3$ (cIP$_3$) induced by UV light pulses of increasing duration (arrows). (**A**) A representative control trace without TPEN, and (**B**) following the addition of 10 µM TPEN between the third and the fourth pulses. The broken line rectangle is magnified in the inset, farthest right side of the panel displaying the near immediate termination of an ongoing rise. (**C, D**) Recordings started in the presence of 10 µM TPEN but in (**D**) 100 µM ZnSO$_4$ was added between the second and the third pulses. (**E**) Recording started in the presence of 100 µM ZnSO$_4$ followed by the addition of 10 µM TPEN between the third and the fourth pulses. (**F, G**) Recording started in the presence of 100 µM MgSO$_4$ (**F**) or 100 µM CaCl$_2$ (**G**) and 10 µM TPEN added as above. Arrowheads above the different panels indicate the time of TPEN or divalent cation addition. (**H**) Bar graphs summarizing the number and percentages of eggs that responded to a given duration of UV pulses under each of the TPEN ± divalent ions.

The online version of this article includes the following figure supplement(s) for figure 3:

**Figure supplement 1.** Overexpression of endoplasmic reticulum (ER) accessory protein ERp44 did not change the Ca$^{2+}$ responses initiated by m*Plcz1* mRNA microinjection, acetylcholine, or SrCl$_2$.

supplement 1), and overexpression of ERp44 modified neither the Ca$^{2+}$ oscillations induced by agonists (*Figure 3—figure supplement 1*) nor the effectiveness of TPEN to block them (data not shown). Thus, TPEN and Zn$^{2+}$ deficiency most likely inhibits Ca$^{2+}$ release by directly interfering with IP$_3$R1 function rather than modifying this particular regulator.

## $Zn^{2+}$ depletion diminishes the ER $Ca^{2+}$ leak and increases $Ca^{2+}$ store content

Our above $cIP_3$ results that TPEN inhibited $IP_3R1$-mediated $Ca^{2+}$ release and interrupted in-progress $Ca^{2+}$ rises despite the presence of high levels of environmental $IP_3$ suggest its actions are probably independent of $IP_3$ binding, agreeing with an earlier report showing that TPEN did not modify $IP_3$'s affinity for the $IP_3R$ (*Richardson and Taylor, 1993*). Additionally, the presence of a $Zn^{2+}$-binding motif near the C-term cytoplasmic domain of the $IP_3R1$'s channel, which is known to influence agonist-induced $IP_3R1$ gating (*Fan et al., 2015*), led us to posit and examine that $Zn^{2+}$ deficiency may be disturbing $Ca^{2+}$ release to the cytosol and out of the ER. To probe this possibility, we queried if pretreatment with TPEN inhibited $Ca^{2+}$ release through $IP_3R1$. We first used thapsigargin (Tg), a sarco-plasmic/ER $Ca^{2+}$ ATPase pump inhibitor (*Thastrup et al., 1990*) that unmasks a constitutive $Ca^{2+}$ leak out of the ER (*Lemos et al., 2021*); in eggs, we have demonstrated it is mediated at least in part by $IP_3R1$ (*Wakai et al., 2019*). Treatment with TPEN for 15 min slowed the Tg-induced $Ca^{2+}$ leak into the cytosol, resulting in delayed and lowered amplitude $Ca^{2+}$ responses (*Figure 4A*; $p<0.05$). To test whether the reduced response to Tg means that TPEN prevented the complete response of Tg, leaving a temporarily increased $Ca^{2+}$ content in the ER, we added the $Ca^{2+}$ ionophore ionomycin (Io), which empties all stores independently of $IP_3Rs$. Io-induced $Ca^{2+}$ responses were 3.3-fold greater in TPEN-treated cells, supporting the view that TPEN interferes with the ER $Ca^{2+}$ leak (*Figure 4A*; $p<0.05$). We further evaluated this concept using in vitro aged eggs that often display reduced $Ca^{2+}$ store content than freshly collected counterparts (*Abbott et al., 1998*). After culturing eggs in the presence or absence of TPEN for 2 hr, we added Io during $Ca^{2+}$ monitoring, which in TPEN-treated eggs induced bigger $Ca^{2+}$ rises than in control eggs (*Figure 4B*; $p<0.05$). We confirmed that this effect was independent of $IP_3R1$ degradation because TPEN did not change $IP_3R1$ reactivity in unfertilized eggs (*Figure 4C*; $p<0.05$).

Next, we used the genetically encoded FRET sensor D1ER (*Palmer et al., 2004*) to assess the TPEN's effect on the ER's relative $Ca^{2+}$ levels changes following the additions of Tg or Ach. TPEN was added 10 min before 10 µM Tg or 50 µM Ach, and we simultaneously monitored changes in cytosolic and intra-ER $Ca^{2+}$ (*Figure 4D and E*). For the first 3 min, the Tg-induced decrease in $Ca^{2+}$-ER was similar between groups. However, while the drop in $Ca^{2+}$ content continued in control eggs, in TPEN-treated eggs, it came to an abrupt halt, generating profound differences between the two groups (*Figure 4D*; $p<0.05$). TPEN had even more pronounced effects following the addition of Ach, leading to a reduced- and prematurely terminated $Ca^{2+}$ release from the ER in treated eggs (*Figure 4E*; $p<0.05$).

Lastly, we sought to use a cellular model where low labile $Zn^{2+}$ occurred without pharmacology. To this end, we examined a genetic model where the two non-selective plasma membrane channels that could influx $Zn^{2+}$ in maturing oocytes have been deleted (*Bernhardt et al., 2017*; *Carvacho et al., 2016*; *Carvacho et al., 2013*), namely, the transient receptor potential melastatin-7 (TRPM7) and TRP vanilloid 3 (TRPV3), both members of the TRP superfamily of channels (*Wu et al., 2010*). We found that eggs from double knockout females (dKOs) had lower levels of labile $Zn^{2+}$ (*Figure 4F*), and the addition of Tg revealed an expanded $Ca^{2+}$ store content in these eggs vs. control WT eggs (*Figure 4G*). Remarkably, in dKO eggs, the $Ca^{2+}$ rise induced by Tg showed a shoulder or inflection point before the peak delaying the time to peak (*Figure 4G*, inset; $p<0.001$). These results in dKO eggs show a changed dynamic of the Tg-induced $Ca^{2+}$ release, suggesting that lower levels of labile $Zn^{2+}$ modify ER $Ca^{2+}$ release independently of chelators.

## $Ca^{2+}$ oscillations in eggs occur within a window of $Zn^{2+}$ concentrations

We next examined whether resupplying $Zn^{2+}$ could restart the $Ca^{2+}$ oscillations terminated by $Zn^{2+}$ depletion. Zn pyrithione (ZnPT) rapidly increases cellular $Zn^{2+}$ upon extracellular addition (*Barnett et al., 1977*; *Robinson, 1964*). Dose titration studies and imaging fluorimetry revealed that 0.01 µM ZnPT caused subtle and protracted increases in $Zn^{2+}$ levels, whereas 0.1 µM ZnPT caused rapid increases in eggs' $Zn^{2+}$ baseline (*Figure 5A*). We induced detectable $Ca^{2+}$ oscillations by injection of m*Plcz1* mRNA followed by 50 µM TPEN (*Figure 5B*), which terminated them. After 30 min, we added 0.1 µM ZnPT, and within 15 min the oscillations restarted in most TPEN-treated eggs (*Figure 5C*). We repeated this approach using thimerosal (*Figure 5D and E*). Adding 0.1 µM ZnPT did not restore the

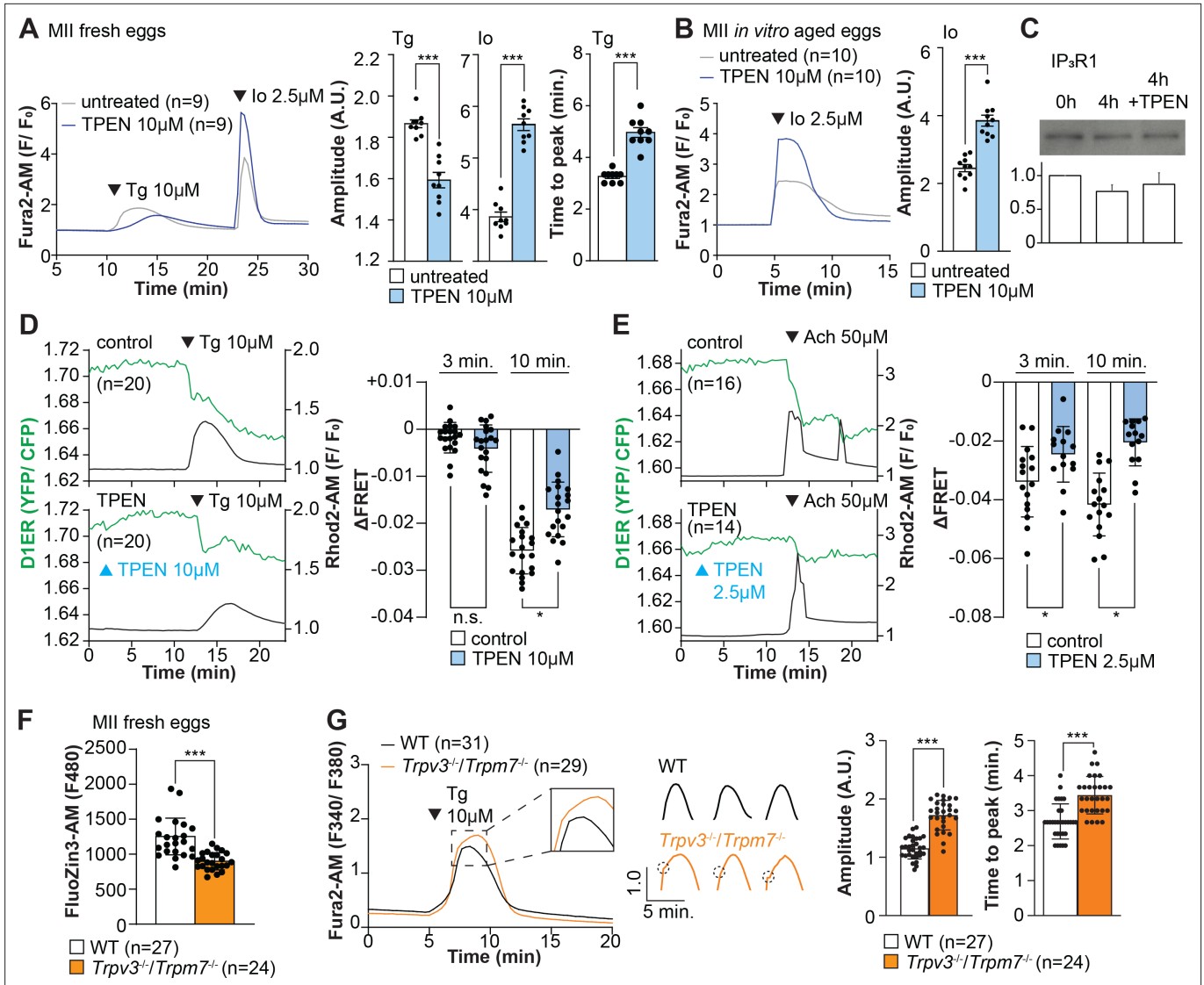

**Figure 4.** Zn²⁺ depletion alters Ca²⁺ homeostasis and increases Ca²⁺ store content independent of IP₃R1 mass. (**A, B**) Representative Ca²⁺ traces of MII eggs after the addition of Tg and Io in the presence or absence of TPEN. Blue trace recordings represent TPEN-treated eggs, whereas gray traces represent control, untreated eggs. (**A**) Io was added to fresh MII eggs once Ca²⁺ returned to baseline after treatment with Tg. Comparisons of mean peak amplitudes after Tg and Io are shown in the bar graphs in the right panel (p<0.001). (**B**) MII eggs were aged by 2 hr. incubation supplemented or not with TPEN followed by Io addition and Ca²⁺ monitoring (p<0.001). (**C**) Western blot showing the intensities of IP₃R1 bands in MII eggs freshly collected, aged by 4 hr. incubation without TPEN, and with TPEN. Thirty eggs per lane in all cases. This experiment was repeated three times. (**D, E**) Left panels: representative traces of Ca²⁺ values in eggs loaded with the Ca²⁺-sensitive dye Rhod-2 AM and the ER Ca²⁺reporter, D1ER (1 µg/µl mRNA). TPEN was added into the media followed 10 min later by (**D**) 10 µM Tg and (**E**) 50 µM Ach. Right panel: bars represent the difference of FRET value between at the time of Tg/ Ach addition and at 3 and 5 min later of the addition (p<0.05). Experiments were repeated two different times for each treatment. Black and green traces represent cytosolic Ca²⁺ and Ca²⁺-ER, respectively. Blue and black arrowheads indicate the time of addition of TPEN and Tg/ Ach, respectively. (**F**) Basal Zn²⁺ level comparison in WT (open bar) and *Trpv3⁻/⁻/Trpm7⁻/⁻* (dKO, orange bar) MII eggs. Each plot represents the Fluozin3 measurement at 5 min after starting monitoring. (**G**) Left panel: representative Ca²⁺ traces of WT (black trace) and dKO (orange trace) MII eggs after adding Tg. Insets represent the magnified traces at the peak of Ca²⁺ spike from different sets of eggs. Middle panel: individual traces of WT and dKO eggs after Tg addition. Dashed circles represent the flection point in dKO traces. Right panel: comparisons of mean peak amplitudes after Tg and the time between Tg addition and the Ca²⁺ peak are shown in the bar graphs in the right panel (p<0.001).

The online version of this article includes the following source data for figure 4:

**Source data 1.** IP₃R1 and TPEN western blotting.

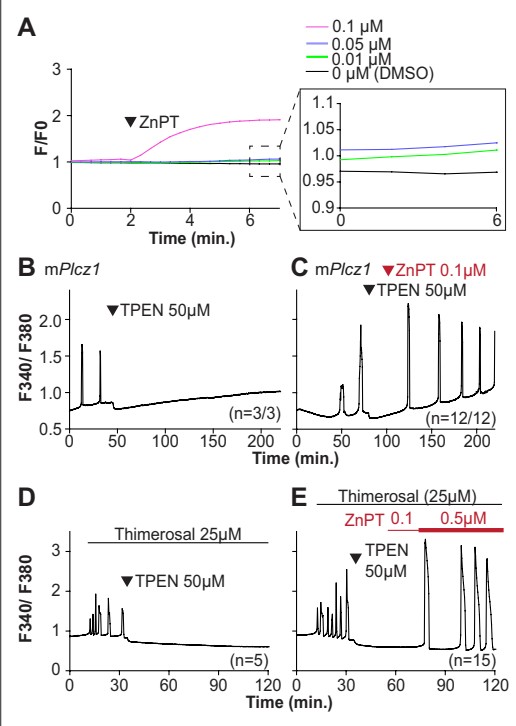

**Figure 5.** Restoring Zn²⁺ levels with ZnPT rescues oscillations interrupted by TPEN-induced Zn²⁺ deficiency. (**A**) Representative traces of Zn²⁺ in MII eggs following the addition of 0.01–0.1 μM concentrations of ZnPT. The broken rectangular area is amplified in the next panel to appreciate the subtle increase in basal Zn²⁺ caused by the addition of ZnPT. (**B, C**) m*Plcz1* mRNA (0.01 μg/μl)-induced oscillations followed by the addition of TPEN (black arrowhead) (**B**), or after the addition of TPEN followed by ZnPT (red arrowhead) (**C**). (**D, E**) Thimerosal (25 μM) induced oscillations using the same sequence of TPEN (**D**) and ZnPT (**E**), but higher concentrations of ZnPT were required to rescue thimerosal-initiated oscillations (**E**). These experiments were repeated at least two different times.

Ca²⁺ oscillations retrained by TPEN, but 0.5 μM ZnPT did so (*Figure 5E*). These results demonstrate that Zn²⁺ plays a pivotal, enabling role in the generation of Ca²⁺ oscillations in mouse eggs.

## Excessive intracellular Zn²⁺ inhibits Ca²⁺ oscillations

Zn²⁺ is necessary for diverse cellular functions, consistent with numerous amino acids and proteins capable of binding Zn²⁺ within specific and physiological ranges (*Pace and Weerapana, 2014*). Excessive Zn²⁺, however, can cause detrimental effects on cells and organisms (*Broun et al., 1990*; *Hara et al., 2022*; *Sikora and Ouagazzal, 2021*). Consistent with the deleterious effects of Zn²⁺, a previous study showed that high concentrations of ZnPT, ~50 μM, prevented SrCl₂-induced egg activation and initiation of development (*Bernhardt et al., 2012*; *Kim et al., 2011*). We examined how ZnPT and excessive Zn²⁺ levels influence Ca²⁺ oscillations. Our conditions revealed that pre-incubation or continuous exposure to 0.1 μM or 1.0 μM ZnPT delayed or prevented egg activation induced by m*Plcz1* mRNA injection (*Figure 6—figure supplement 1*). We used these ZnPT concentrations to add it into ongoing oscillations induced by ICSI and monitored the succeeding Ca²⁺ responses. The addition of 0.05–10 μM ZnPT caused an immediate elevation of the basal levels of Fura-2 and termination of the Ca²⁺ oscillations (*Figure 6A–D*). m*Plcz1* mRNA-initiated Ca²⁺ responses were also interrupted by adding 0.1 μM ZnPT, whereas untreated eggs continued oscillating (*Figure 6E and F*). ZnPT also inhibited IP₃R1-mediated Ca²⁺ release triggered by cIP₃, suggesting that excessive Zn²⁺ directly inhibits IP₃R1 function (*Figure 6G*).

A noticeable feature of ZnPT is the increased basal ratios of Fura-2 AM. These changes could reflect enhanced IP₃R1 function and increased basal Ca²⁺ concentrations caused by Zn²⁺ stimulation of IP₃R1. This seems unlikely, however, because extended elevated cytosolic Ca²⁺ would probably induce cellular responses, such as the release of the second polar body, egg fragmentation, or cell death, neither of which happened. It might reflect, instead, Fura-2's ability to report changes in Zn²⁺ levels, which seemed the case because the addition of TPEN lowered fluorescence without restarting the Ca²⁺ oscillations (*Figure 6F*). To ensure the impact of ZnPT abolishing Ca²⁺ oscillations was not an imaging artifact obscuring ongoing rises, we simultaneously monitored cytoplasmic and ER Ca²⁺ levels with Rhod-2 and D1ER, respectively. This approach allowed synchronously observing Ca²⁺ changes in both compartments that should unfold in opposite directions. In control, uninjected eggs, the fluorescent values for both reporters remained unchanged during the monitoring period, whereas in m*Plcz1* mRNA-injected eggs, the reporters' signals displayed simultaneous but opposite changes, as expected (*Figure 6H and I*). The addition of ZnPT in uninjected eggs rapidly increased Rhod-2 signals but not D1ER's, which was also the case in oscillating eggs, as the addition of ZnPT did not immediately alter the dynamics of the ER's Ca²⁺ release, suggesting that D1ER faithfully reports in Ca²⁺ changes but cannot detect changes in Zn²⁺ levels, at least to this extent; ZnPT progressively caused fewer and lower amplitude changes

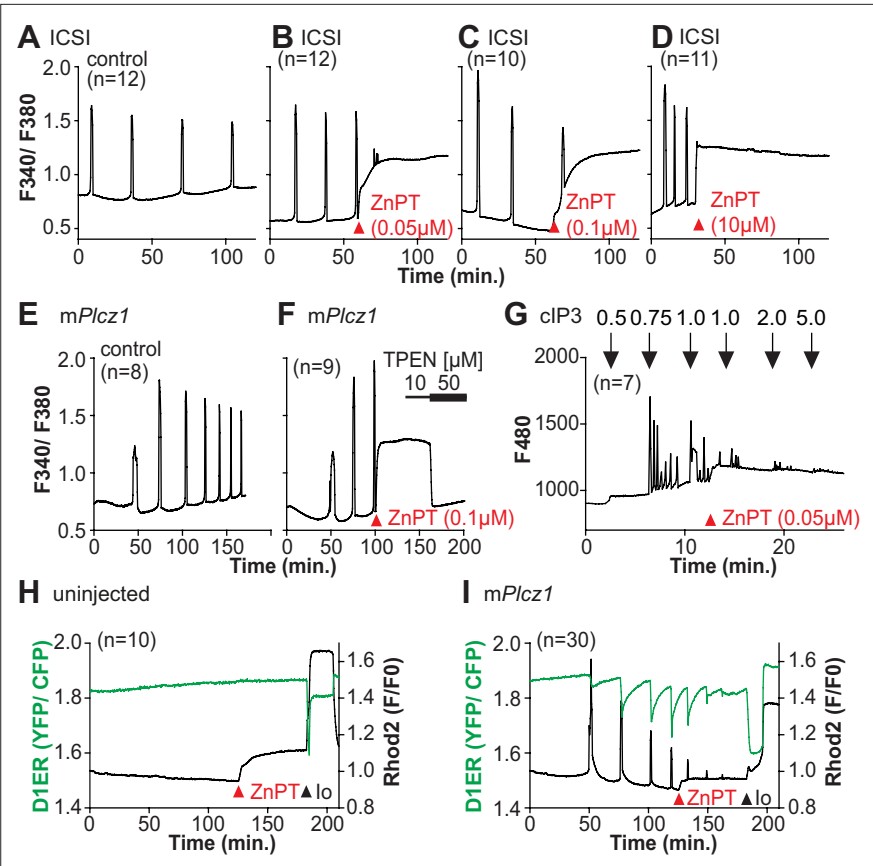

**Figure 6.** Excess Zn²⁺ hinders Ca²⁺ oscillations. (**A–D**) Intracytoplasmic sperm injection (ICSI)-initiated Ca²⁺ response without (**A**) or following the addition of ZnPT (**B, C**) (the time of ZnPT addition and concentration are denoted above the tracing). (**E, F**) Representative Ca²⁺ responses induced by injection of 0.01 µg/µl m*Plcz1* mRNA in untreated eggs (**E**) or in eggs treated with 0.1 µM ZnPT followed by 10 µM TPEN first and then 50 µM (**F**). (**G**) cIP₃-induced Ca²⁺ release as expected when the UV pulses in the absence but not in the presence of 0.05 µM ZnPT (the time of addition is denoted by a bar above the tracing). (**H, I**) Representative traces of Ca²⁺ values in eggs loaded with the Ca²⁺-sensitive dye Rhod-2 AM and the ER Ca²⁺ reporter, D1ER (1 µg/µl mRNA). Uninjected and 0.01 µg/µl m*Plcz1* mRNA-injected eggs were monitored. After initiation and establishment of the oscillations, 0.1 µM ZnPT was added into the media followed 30 min later by 2.5 µM Io. Experiments were repeated two different times. Red and black arrowheads indicate the time of addition of ZnPT and Io, respectively.

The online version of this article includes the following figure supplement(s) for figure 6:

**Figure supplement 1.** Elevated Zn²⁺ impairs egg activation and the subsequent embryo development.

in D1ER fluorescence, consistent with the diminishing and eventual termination of the Ca²⁺ oscillations. Noteworthy, in these eggs, the basal D1ER fluorescent ratio remained unchanged after ZnPT, demonstrating its unresponsiveness to Zn²⁺ changes of this magnitude. The ZnPT-induced increases in Rhod-2 fluorescence without concomitant changes in D1ER values suggest that the changes in the dyes' fluorescence do not represent an increase in basal Ca²⁺ and, more likely, signal an increase in intracellular Zn²⁺. We confirmed that both reporters were still in working order as the addition of Io triggered Ca²⁺ changes detected by both reporters (*Figure 6H and I*).

## Discussion

This study demonstrates that appropriate levels of labile Zn²⁺ are essential for initiating and maintaining IP₃R1-mediated Ca²⁺ oscillations in mouse eggs regardless of the initiating stimuli. Both deficient and excessive Zn²⁺ compromise IP₃R1 sensitivity, diminishing and mostly terminating Ca²⁺ oscillations. The results demonstrate that IP₃R1 and Zn²⁺ act in concert to modulate Ca²⁺ signals, revealing previously unexplored crosstalk between these ions at fertilization (*Figure 7*).

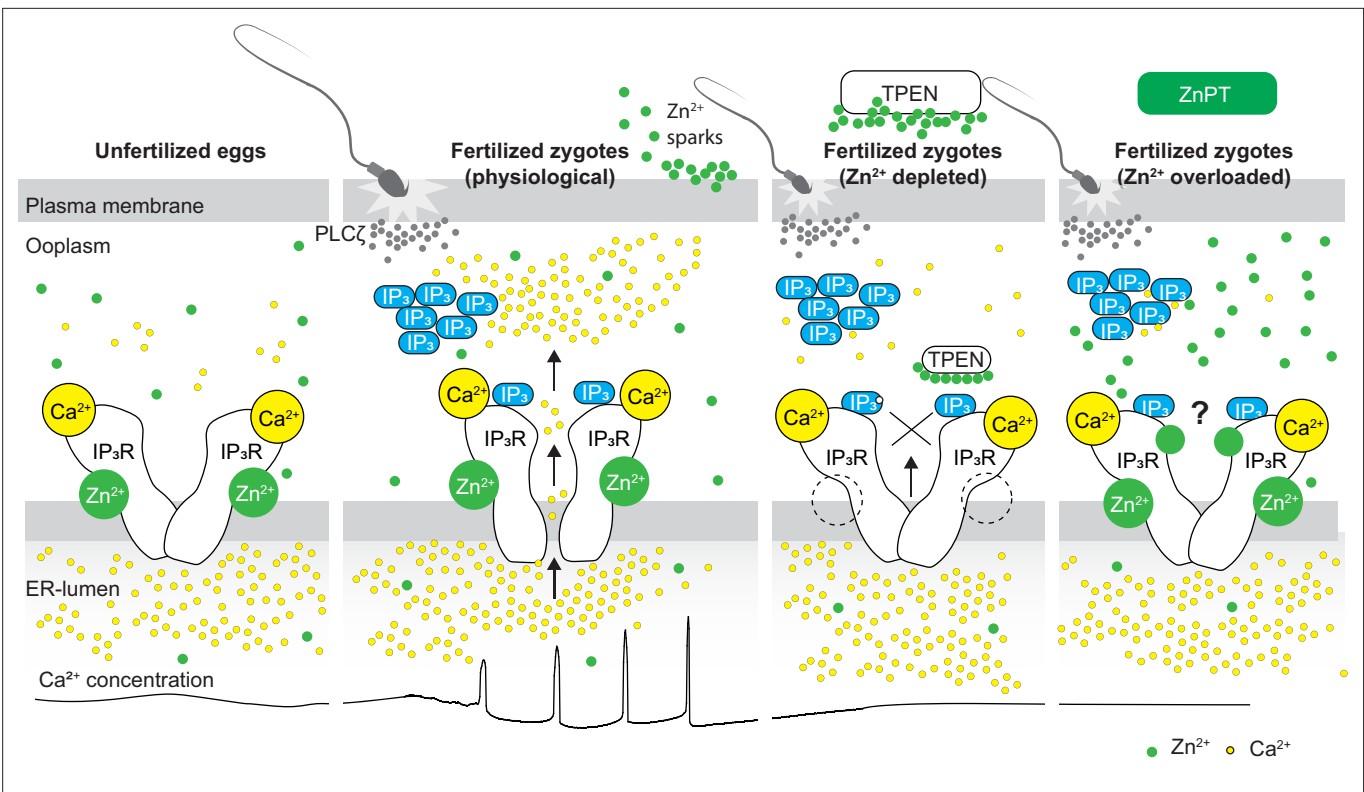

**Figure 7.** Schematic of proposed regulation of IP₃R1 function by $Zn^{2+}$ in eggs and fertilized zygotes. In MII eggs, left panel, IP₃R1s are in a $Ca^{2+}$-release permissive state with optimal levels of cytoplasmic $Ca^{2+}$ and $Zn^{2+}$ and maximum endoplasmic reticulum (ER) content, but $Ca^{2+}$ is maintained at resting levels by the combined actions of pumps, ER $Ca^{2+}$ leak, and reduced influx. Once fertilization takes place, left center panel, robust IP₃ production induced by the sperm-borne PLC ζ leads to $Ca^{2+}$ release through ligand-induced gating of IP₃R1. Continuous IP₃ production and refilling of the stores via $Ca^{2+}$ influx ensure the persistence of the oscillations. $Zn^{2+}$ release occurs in association with first few $Ca^{2+}$ rises and cortical granule exocytosis, $Zn^{2+}$ sparks, lowering $Zn^{2+}$ levels but not sufficiently to inhibit IP₃R1 function. $Zn^{2+}$ deficiency caused by TPEN or other permeable $Zn^{2+}$ chelators, right center panel, dose-dependently impairs IP₃R1 function and limits $Ca^{2+}$ release. We propose this is accomplished by stripping the $Zn^{2+}$ bound to the residues of the zinc-finger motif in the LNK domain of IP₃R1 that prevents the allosteric modulation of the gating process induced by IP₃ or other agonists. We propose that excess $Zn^{2+}$, right panel, also inhibits IP₃R1-mediate $Ca^{2+}$ release, possibly by non-specific binding of thiol groups present in cysteine residues throughout the receptor (denoted by a?). We submit that optimal $Ca^{2+}$ oscillations in mouse eggs unfold in the presence of a permissive range of $Zn^{2+}$ concentration.

$Zn^{2+}$ is an essential micronutrient for living organisms (**Kaur et al., 2014**) and is required for various cellular functions, such as proliferation, transcription, and metabolism (**Lo et al., 2020**; **Maret and Li, 2009**; **Yamasaki et al., 2007**). Studies using $Zn^{2+}$ chelators have uncovered what appears to be a cell-specific, narrow window of $Zn^{2+}$ concentrations needed for cellular proliferation and survival (**Carraway and Dobner, 2012**; **Lo et al., 2020**). Further, TPEN appeared especially harmful, and in a few cell lines, even low doses provoked oxidative stress, DNA fragmentation, and apoptosis (**Mendivil-Perez et al., 2012**). We show here that none of the $Zn^{2+}$ chelators, permeable or impermeable, affected cell viability within our experimental observations, confirming findings from previous studies that employed high concentrations of TPEN to interrupt the $Ca^{2+}$ oscillations (**Lawrence et al., 1998**) or inducing egg activation of mouse eggs (**Suzuki et al., 2010b**). Our data demonstrating that ~2.5 µM is the threshold concentration of TPEN in eggs that first causes noticeable changes in basal $Zn^{2+}$, as revealed by FluoZin, is consistent with the ~2–5 µM $Zn^{2+}$ concentrations in most culture media without serum supplementation (**Lo et al., 2020**), and with the ~100 pM basal $Zn^{2+}$ in cells (**Qin et al., 2011**). Lastly, the effects on $Ca^{2+}$ release observed here with TPEN and other chelators were due to the chelation of $Zn^{2+}$, as pretreatment with $ZnSO_4$ but not with equal or greater concentrations of $MgCl_2$ or $CaCl_2$ rescued the inhibition of the responses, which is consistent with results by others (**Kim et al., 2010**; **Lawrence et al., 1998**).

To identify how $Zn^{2+}$ deficiency inhibits $Ca^{2+}$ release in eggs, we induced $Ca^{2+}$ oscillations using various stimuli and tested the effectiveness of membrane-permeable and -impermeable chelators to abrogate them. Chelation of extracellular $Zn^{2+}$ failed to terminate the $Ca^{2+}$ responses, whereas membrane-permeable chelators did, pointing to intracellular labile $Zn^{2+}$ levels as essential for $Ca^{2+}$ release. All agonists used here were susceptible to inhibition by TPEN, whether their activities depended on $IP_3$ production or allosterically induced receptor function, although the effective TPEN concentrations varied across stimuli. Some agents, such as m*Plcz1* mRNA or thimerosal, required higher concentrations than $SrCl_2$, Ach, or $cIP_3$. The reason underlying the different agonists' sensitivities to TPEN will require additional research, but the persistence of $IP_3$ production or change in $IP_3R1$ structure needed to induce channel gating might explain it. However, the universal abrogation of $Ca^{2+}$ oscillations by TPEN supports the view drawn from cryo-EM-derived $IP_3R1$ models that signaling molecules can allosterically induce channel gating from different starting positions in the receptor by mechanically coupling the binding effect to the ion-conducting pore in the C-terminal end of $IP_3R$ (*Fan et al., 2015*). The cytosolic C-terminal domain of each $IP_3R1$ subunit is alongside the $IP_3$-binding domain of another subunit and, therefore, well positioned to sense $IP_3$ binding and induce channel gating (*Fan et al., 2015*). Within each subunit, the LNK domain, which contains a $Zn^{2+}$-finger motif (*Fan et al., 2015*), connects the opposite domains of the molecule. Although there are no reports regarding the regulation of $IP_3R1$ sensitivity by $Zn^{2+}$, such evidence exists for RyRs (*Woodier et al., 2015*), which also display a conserved $Zn^{2+}$-finger motif (*des Georges et al., 2016*). Lastly, mutations of the two Cys or two His residues of this motif, without exception, resulted in inhibition or inactivation of the $IP_3R1$ channel (*Bhanumathy et al., 2012*; *Uchida et al., 2003*). These results are consistent with the view that the C-terminal end of $IP_3Rs$ plays a dominant role in channel gating (*Bhanumathy et al., 2012*; *Uchida et al., 2003*). We propose that TPEN inhibits $Ca^{2+}$ oscillations in mouse eggs because chelating $Zn^{2+}$ interferes with the function of the LNK domain and its $Zn^{2+}$-finger motif proposed role on the mechanical coupling induced by agonist binding to the receptor that propagates to the pore-forming region and required to gate the channel's ion-pore (*Fan et al., 2022*; *Fan et al., 2015*).

In support of this possibility, TPEN-induced $Zn^{2+}$-deficient conditions altered the $Ca^{2+}$-releasing kinetics in resting eggs or after fertilization. Tg increases intracellular $Ca^{2+}$ by inhibiting the SERCA pump (*Thastrup et al., 1990*) and preventing the reuptake into the ER of the ebbing $Ca^{2+}$ during the basal leak. Our previous studies showed that the downregulation of $IP_3R1$ diminishes the leak, suggesting that it occurs through $IP_3R1$ (*Wakai and Fissore, 2019*). Consistent with this view, TPEN pretreatment delayed the $Ca^{2+}$ response induced by Tg, implying that $Zn^{2+}$ deficiency hinders $Ca^{2+}$ release through $IP_3R1$. An expected consequence would be increased $Ca^{2+}$ content in the ER after Tg. Io that mobilizes $Ca^{2+}$ independently of $IP_3Rs$ (*Toeplitz et al., 1979*) induced enhanced responses in TPEN-treated eggs vs. controls, confirming the accumulation of $Ca^{2+}$- ER in $Zn^{2+}$-deficient conditions. We demonstrated that this accumulation is due to hindered emptying of the $Ca^{2+}$ ER evoked by agonists in $Zn^{2+}$-deficient environments, resulting in reduced cytosolic $Ca^{2+}$ increases, as $IP_3R1$ is the pivotal intermediary channel between these compartments. Noteworthy, the initial phase of the Tg-induced $Ca^{2+}$ release out of the ER did not appear modified by TPEN, as if it was mediated by a $Zn^{2+}$-insensitive $Ca^{2+}$ channel(s)/transporter, contrasting with the abrogation of Ach-induced ER emptying from the outset. Remarkably, independently of $Zn^{2+}$ chelators, emptying of $Ca^{2+}$ ER was modified in a genetic model of $Zn^{2+}$-deficient oocytes lacking two TRP channels, confirming the impact of $Zn^{2+}$ on $Ca^{2+}$ release. It is worth noting that TPEN did not reduce but maintained or increased the mass of $IP_3R1$, which might result in the inhibition of $Zn^{2+}$-dependent ubiquitin ligase Ubc7 by the Zn-deficient conditions (*Webster et al., 2003*). We cannot rule out that these conditions may undermine other conformational changes required to trigger $IP_3R1$ degradation, thereby favoring the accumulation of $IP_3R1$.

Despite accruing $Zn^{2+}$ during oocyte maturation, fertilization witnesses a necessary $Zn^{2+}$ release into the external milieu, known as '$Zn^{2+}$ sparks' (*Converse and Thomas, 2020*; *Kim et al., 2011*; *Mendoza et al., 2022*; *Que et al., 2019*; *Que et al., 2015*; *Seeler et al., 2021*). This release of $Zn^{2+}$ is a conserved event in fertilization across species and is associated with several biological functions, including those related to fending off polyspermy (*Kim et al., 2011*; *Que et al., 2019*; *Wozniak et al., 2020*). The concomitant decrease in $Zn^{2+}$ facilitates the resumption of the cell cycle and exit from the MII stage (*Kim et al., 2011*). Congruent with this observation, artificial manipulation that maintains high $Zn^{2+}$ levels prevents egg activation (*Kim et al., 2011*), whereas lowering $Zn^{2+}$ with

chelators leads to egg activation without $Ca^{2+}$ mobilization (*Suzuki et al., 2010b*). As posed by others, these results suggest that meiosis completion and the early stages of fertilization unfold within a narrow window of permissible $Zn^{2+}$ (*Kim et al., 2011*; *Kim et al., 2010*). Here, we extend this concept and show that $IP_3R1$ function and the $Ca^{2+}$ oscillations in mouse eggs require this optimal level of labile $Zn^{2+}$ because the $Ca^{2+}$ responses interrupted by TPEN-induced $Zn^{2+}$-insufficiency are rescued by restoring $Zn^{2+}$ levels with ZnPT. Furthermore, unopposed increases in $Zn^{2+}$ by exposure to ZnPT abrogated fertilization-initiated $Ca^{2+}$ oscillations and prevented the expected egg activation events. It is unclear how excess $Zn^{2+}$ disturbs the function of $IP_3R1$. Nevertheless, $IP_3R1$s have multiple cysteines whose oxidation enhances the receptor sensitivity to $IP_3$ (*Joseph et al., 2018*), and it is possible that excessive $Zn^{2+}$ aberrantly modifies them, disturbing $IP_3R1$ structure and function or, alternatively, preventing their oxidation and sensitization of the receptor. Lastly, we cannot rule out that high $Zn^{2+}$ levels directly inhibit the receptor's channel. These results reveal a close association between the $Zn^{2+}$ levels controlling meiotic transitions and the $Ca^{2+}$ release necessary for egg activation, placing the $IP_3R1$ at the center of the crosstalk of these two divalent cations.

Abrupt $Zn^{2+}$ changes have emerged as critical signals for meiotic and mitotic transitions in oocytes, eggs, embryos, and somatic cells (*Kim et al., 2011*; *Kim et al., 2010*; *Lo et al., 2020*). Fertilization relies on prototypical $Ca^{2+}$ rises and oscillations, and $Zn^{2+}$ sparks are an egg activation event downstream of this $Ca^{2+}$ release, establishing a functional association between these two divalent cations that continues to grow (*Kim et al., 2011*). Here, we show that, in addition, these cations actively crosstalk during fertilization and that the fertilization-induced $Ca^{2+}$ oscillations rely on optimized $IP_3R1$ function underpinned by ideal $Zn^{2+}$ levels set during oocyte maturation. Future studies should explore if artificial alteration of $Zn^{2+}$ levels can extend the fertile lifespan of eggs, improve developmental competence, or develop methods of non-hormonal contraception.

# Materials and methods

**Key resources table**

| Reagent type (species) or resource | Designation | Source or reference | Identifiers | Additional information |
|---|---|---|---|---|
| Genetic reagent (*Mus musculus*) | CD1 | Charles River | 022 | |
| Genetic reagent (*M. musculus*) | C57BL/6J | JAX | JAX: 000664 | |
| Genetic reagent (*M. musculus*) | *Trpm7*-floxed | A generous gift from Dr. Carmen P. Williams (NIEHS) (PMID:30322909) | | C57BL6/J and 129s4/SvJae mixed background |
| Genetic reagent (*M. musculus*) | *Gdf9-cre* | JAX | JAX: 011062 | |
| Genetic reagent (*M. musculus*) | *Trpv3*$^{-/-}$ | A generous gift from Dr H. Xu (PMID:20403327) | | C57BL/6J and 129/SvEv mixed background |
| Biological sample (mouse oocyte) | *Mus musculus* | This paper | | Eggs at the metaphase of the second meiosis |
| Biological sample (mouse sperm) | *Mus musculus* | This paper | | Matured sperm from cauda epididymis |
| Recombinant DNA reagent | pcDNA6-mouse *Plcz1-venus* (plasmid used as a template for mRNA synthesis) | Published in previous Fissore lab paper PMID: 34313315 Mouse *Plcz1* sequence was a generous gift from Dr. Kiyoko Fukami (PMID:18028898) | | Mouse *Plcz1* mRNA was fused with Venus and inserted in pcDNA6 vector |
| Recombinant DNA reagent | pcDNA6-CALR-D1ER-KDEL (plasmid used as a template for mRNA synthesis) | Published in previous Fissore lab paper PMID:24101727 Original D1ER vector was a generous gift from Dr. Roger Y Tsien (PMID:15585581) | | FRET construct D1ER was inserted between ER-targeting sequence of calreticulin and KDEL ER retention signal in pcDNA6 vector |

*Continued on next page*

*Continued*

| Reagent type (species) or resource | Designation | Source or reference | Identifiers | Additional information |
|---|---|---|---|---|
| Recombinant DNA reagent | pcDNA6-human *ERP44-HA* (plasmid used as a template for mRNA synthesis) | This paper Original human ERp44 sequence was a generous gift from Dr. Roberto Sitia (PMID:11847130) | | Human *ERP44* mRNA fused with HA in pcDNA6/Myc His B vector |
| Antibody | Monoclonal HA (mouse monoclonal) | Roche | 11581816001 | Dilution: 1:200 |
| Antibody | Polyclonal IP$_3$R1 (rabbit polyclonal) | *Parys et al., 1995* | | Dilution: 1:1000 |
| Antibody | Monoclonal α-tubulin (mouse monoclonal) | Sigma-Aldrich | T-9026 | Dilution: 1:1000 |
| Antibody | Alexa Fluor 488 (goat polyclonal) | Invitrogen | Invitrogen: A32723 | Dilution: 1:400 |
| Commercial assay or kit | T7 mMESSAGE mMACHINE Kit | Invitrogen | Invitrogen: AM1344 | Used for in vitro mRNA synthesis |
| Commercial assay or kit | Poly(A) Tailing Kit | Invitrogen | Invitrogen: AM1350 | Used for poly (A) tailing of synthesized mRNA |
| Chemical compound, drug | Hyaluronidase from bovine testes | Sigma-Aldrich | H3506 | |
| Chemical compound, drug | 3-Isobutyl-1-methylxanthine (IBMX) | Sigma-Aldrich | I5879 | |
| Chemical compound, drug | Polyvinylpyrrolidone (PVP) (average molecular weight: 360,000) | Sigma-Aldrich | PVP360 | Used for mRNA microinjection and ICSI |
| Chemical compound, drug | N,N, N′,N′-Tetrakis (2-pyridylmethyl) ethylenediamine (TPEN) | Sigma-Aldrich | P4413 | Prepared in DMSO and kept at –20°C until use |
| Chemical compound, drug | Zinc pyrithione (ZnPT) | Sigma-Aldrich | PHR1401 | Prepared in DMSO and kept at –20°C until use |
| Chemical compound, drug | Strontium chloride hexahydrate (SrCl$_2$) | Sigma-Aldrich | 255521 | Freshly dissolved in water on the day of experiment |
| Chemical compound, drug | Calcium chloride dihydrate (CaCl$_2$) | Sigma-Aldrich | C3881 | Freshly dissolved in water on the day of experiment |
| Chemical compound, drug | Magnesium chloride hexahydrate (MgCl$_2$) | Sigma-Aldrich | M2393 | Freshly dissolved in water on the day of experiment |
| Chemical compound, drug | Zinc sulfate monohydrate (ZnSO$_4$) | Acros Organics | 389802500 | Freshly dissolved in water on the day of experiment |
| Chemical compound, drug | Ethylenediaminetetraacetic acid sodium dihydrate (EDTA) | LabChem | LC137501 | Prepared as 0.5 M aqueous solution with pH 8.0 adjusted by NaOH |
| Chemical compound, drug | Diethylenetriaminepentaacetic acid (DTPA) | Sigma-Aldrich | D6518 | |
| Chemical compound, drug | Tris (2-pyridylmethyl) amine (TPA) | Santa Cruz | sc-477037 | |
| Chemical compound, drug | Dimethyl sulfoxide (DMSO) | Sigma-Aldrich | D8418 | Used as a solvent |
| Chemical compound, drug | Acetylcholine chloride | Sigma-Aldrich | A6625 | |
| Chemical compound, drug | Thimerosal | Sigma-Aldrich | T5125 | Freshly dissolved in water on the day of experiment and kept on ice until use |

*Continued*

| Reagent type (species) or resource | Designation | Source or reference | Identifiers | Additional information |
|---|---|---|---|---|
| Chemical compound, drug | Ionomycin calcium salt | Tocris | 1704 | Working concentration: 2.5 µM |
| Chemical compound, drug | Thapsigargin | Calbiochem | #586500 | Working concentration: 10 µM |
| Other | Pluronic F-127 (20% solution in DMSO) (pluronic acid) | Invitrogen | P3000MP | Added to dye dilutions to facilitate the solubilization |
| Other | Fura-2 AM | Invitrogen | F1221 | Ratiometric fluorescent $Ca^{2+}$ indicator Used at 1.25 µM in TL-HEPES containing 0.02% pluronic acid |
| Other | FluoZin-3 AM | Invitrogen | F24195 | Fluorescent $Zn^{2+}$ indicator Used at 1.25 µM in TL-HEPES containing 0.02% pluronic acid |
| Other | Fluo-4 AM | Invitrogen | F14201 | Fluorescent $Ca^{2+}$ indicator Used at 1.25 µM in TL-HEPES containing 0.02% pluronic acid |
| Other | Rhod2-AM | Invitrogen | R1244 | Fluorescent $Ca^{2+}$ indicator Used at 2.2 µM in TL-HEPES containing 0.02% pluronic acid. |
| Other | ci-IP3/PM | Tocris | 6210 | Photo-activatable $IP_3$. Dissolved in DMSO and kept at –20°C Before use, the stock was diluted with water to make a final concentration of 0.25 mM |
| Other | Pme1 | New England BioLabs | R0560S | Used to linearize pcDNA6 vectors for mRNA synthesis |
| Software, algorithm | Prism | GraphPad Software | | Version 5.01 |

N,N,N′,N′-tetrakis (2-pyridinylmethyl)–1,2-ethylenediamine (TPEN) and zinc pyrithione (ZnPT) were dissolved in dimethyl sulfoxide (DMSO) at 10 mM and stored at –20°C until use. $SrCl_2$, $CaCl_2$, $ZnSO_4$, and $MgCl_2$ were freshly dissolved with double-sterile water at 1 M and diluted with the monitoring media just before use. Ethylenediaminetetraacetic acid (EDTA) and diethylenetriaminepentaacetic acid (DTPA) were reconstituted with double-sterile water at 0.5 M and 10 mM, respectively, and the pH was adjusted to 8.0. Tris(2-pyridylmethyl) amine (TPA) was diluted in DMSO at 100 mM and stored at –20°C until use. Acetylcholine chloride and thimerosal were dissolved in double-sterile water at 550 mM and 100 mM, respectively. Acetylcholine was stored at –20°C until use, whereas thimerosal was made fresh in each experiment.

## Mice

The University of Massachusetts Institutional Animal Care and Use Committee (IACUC) approved all animal experiments and protocols. *Trpm7*-floxed (*Trpm7*<sup>fl/fl</sup>) *Gdf9-Cre* and *Trpv3*<sup>–/–</sup> mice were bred at our facility. *Trpm7*<sup>fl/fl</sup> mice were crossed with *Trpv3*<sup>–/–</sup> to generate *Trpm7*<sup>fl/fl</sup>; *Trpv3*<sup>–/–</sup> mouse line. Female *Trpm7*<sup>fl/fl</sup>; *Trpv3*<sup>–/–</sup> mice were crossed with *Trpm7*<sup>fl/fl</sup>; *Trpv3*<sup>–/–</sup>; *Gdf9-cre* male to generate females null for *Trpv3* and with oocyte-specific deletion for *Trpm7*. Ear clips from offspring were collected prior to weaning, and confirmation of genotype was performed after most experiments.

## Egg collection

All gamete handling procedures are as previously reported by us (*Wakai et al., 2019*). MII eggs were collected from the ampulla of 6- to 8-week-old female mice. Females were superovulated via intraperitoneal injections of 5 IU pregnant mare serum gonadotropin (PMSG, Sigma, St. Louis, MO) and 5 IU human chorionic gonadotropin (hCG, sigma) at 48 hr interval. Cumulus-oocyte-complexes (COCs) were obtained 13.5 hr post-hCG injection by tearing the ampulla using forceps and needles in TL-HEPES medium. COCs were treated with 0.26% (w/v) of hyaluronidase at room temperature (RT) for 5 min to remove cumulus cells.

## Intracytoplasmic sperm injection (ICSI)

ICSI was performed as previously reported by us (*Kurokawa and Fissore, 2003*) using described setup and micromanipulators (Narishige, Japan). Sperm from C57BL/6 or CD1 male mice (7–12 weeks old) were collected from the cauda epididymis in TL-HEPES medium, washed several times, heads separated from tails by sonication (XL2020; Heat Systems Inc, USA) for 5 s at 4°C. The sperm lysate was washed in TL-HEPES and diluted with 12% polyvinylpyrrolidone (PVP, MW = 360 kDa) to a final PVP concentration of 6%. A piezo micropipette-driving unit was used to deliver the sperm into the ooplasm (Primetech, Ibaraki, Japan); a few piezo-pulses were applied to puncture the eggs' plasma membrane following penetration of the zona pellucida. After ICSI, eggs were either used for $Ca^{2+}$ monitoring or cultured in KSOM to evaluate activation and development at 36.5°C in a humidified atmosphere containing 5% $CO_2$.

## Preparation and microinjection of mRNA

pcDNA6-m*Plcz1-mEGFP*, pcDNA6-CALR-D1ER-KDEL, and pcDNA6-*humanERp44-HA* were linearized with the restriction enzyme PmeI and in vitro transcribed using the T7 mMESSAGE mMACHINE Kit following procedures previously used in our laboratory (*Ardestani et al., 2020*). A poly(A) tail was added to the in vitro synthesized RNA (mRNA) using Tailing Kit followed by quantification and dilution to 0.5 µg/µl in nuclease-free water and stored at –80°C until use. Before microinjection, m*Plcz1*, D1ER, and *ERp44* mRNA were diluted to 0.01, 1.0, and 0.5 µg/µl, respectively, in nuclease-free water, heated at 95°C for 3 min followed by centrifugation at 13,400 × *g* for 10 min at 4°C. Cytoplasm injection of mRNA was performed under microscopy equipped with micromanipulators (Narishige, Japan). The zona pellucida and the plasma membrane of MII eggs were penetrated by applying small pulses generated by the piezo micromanipulator (Primetech). The preparation of the injection pipette was as for ICSI (*Kurokawa and Fissore, 2003*), but the diameter of the tip was ~1 µm.

## $Ca^{2+}$ and $Zn^{2+}$ imaging

Before $Ca^{2+}$ imaging, eggs were incubated in TL-HEPES containing 1.25 µM Fura2-AM, 1.25 µM FluoZin3-AM, or 2.2 µM Rhod2-AM and 0.02% pluronic acid for 20 min at RT and then washed. The fluorescent probe-loaded eggs were allowed to attach to the bottom of the glass dish (Mat-Tek Corp., Ashland, MA). Eggs were monitored simultaneously using an inverted microscope (Nikon, Melville, NY) outfitted for fluorescence measurements. Fura-2 AM, FluoZin3-AM, and Rhod2-AM fluorescence were excited with 340 nm and 380 nm, 480 nm, and 550 nm wavelengths, respectively, every 20 s, for different intervals according to the experimental design and as previously performed in the laboratory. The illumination was provided by a 75 W Xenon arc lamp and controlled by a filter wheel (Ludl Electronic Products Ltd, Hawthorne, NY). The emitted light above 510 nm was collected by a cooled Photometrics SenSys CCD camera (Roper Scientific, Tucson, AZ). Nikon Element software coordinated the filter wheel and data acquisition. The acquired data were saved and analyzed using Microsoft Excel and GraphPad using Prism software (*Ardestani et al., 2020*). For *Figures 1A, 4A–C, 5A, and 6H–I*, values obtained from FluoZin3-AM, Fura2-AM, or Rhod2-AM recordings were divided by the average of the first five recordings for each treatment that was used as the $F_0$.

To estimate relative changes in $Ca^{2+}$-ER, emission ratio imaging of the D1ER (YFP/CFP) was performed using a CFP excitation filter, dichroic beamsplitter, CFP and YFP emission filters (Chroma Technology, Rockingham, VT; ET436/20X, 89007bs, ET480/40m, and ET535/30m). To measure $Ca^{2+}$-ER and cytosolic $Ca^{2+}$ simultaneously, eggs that had been injected with D1ER were loaded with Rhod-2AM, and CFP, YFP, and Rhod-2 intensities were collected every 20 s.

## Caged $IP_3$

Caged-$IP_3$/PM (c$IP_3$) was reconstituted in DMSO and stored at –20°C until use. Before injection, c$IP_3$ stock was diluted to 0.25 mM with water and microinjected as above. After incubation in KSOM media at 37°C for 1 hr, the injected eggs were loaded with the fluorophore, 1.25 µM Fluo4-AM, and 0.02% pluronic acid and handled as above for Fura-2 AM. The release of c$IP_3$ was accomplished by photolysis using 0.5–5 s pulses at 360 nm wavelengths. $Ca^{2+}$ imaging was as above, but Fluo4 was excited at 488 nm wavelength and emitted light above 510 nm collected as above.

## Western blot analysis

Cell lysates from 20 to 50 mouse eggs were prepared by adding 2× Laemmli sample buffer. Proteins were separated on 5% SDS–PAGE gels and transferred to PVDF membranes (Millipore, Bedford, MA).

After blocking with 5% fat-free milk + TBS, membranes were probed with the rabbit polyclonal antibody specific to $IP_3R1$ (1:1000; a generous gift from Dr. Jan Parys, Katholieke Universiteit, Leuven, Belgium; *Parys et al., 1995*). Goat anti-rabbit antibody conjugated to horseradish peroxidase (HRP) was used as a secondary antibody (1:5000; goat anti-rabbit IgG [H+L] Cross-Adsorbed Secondary Antibody, HRP; Invitrogen, Waltham, MA). For detection of chemiluminescence, membranes were developed using ECL Prime (Sigma) and exposed for 1–3 min to maximum sensitivity film (VWR, Radnor, PA). Broad-range pre-stained SDS–PAGE molecular weight markers (Bio-Rad, Hercules, CA) were run in parallel to estimate the molecular weight of the immunoreactive bands. The same membranes were stripped at 50°C for 30 min (62.5 mM Tris, 2% SDS, and 100 mM 2-beta mercaptoethanol) and re-probed with anti-α-tubulin monoclonal antibody (1:1000).

## Immunostaining and confocal microscopy

Immunostaining was performed according to our previous study (*Akizawa et al., 2021*). After incubation with or without TPEN, MII eggs were fixed with 4% (w/v) paraformaldehyde in house-made phosphate-buffered saline (PBS) for 20 min at RT and then permeabilized for 60 min with 0.2% (v/v) Triton X-100 in PBS. Next, the eggs were blocked for 45 min with a blocking buffer containing 0.2% (w/v) skim milk, 2% (v/v) fetal bovine serum, 1% (w/v) bovine serum albumin, 0.1% (v/v) TritonX-100, and 0.75% (w/v) glycine in PBS. Eggs were incubated overnight at 4°C with mouse anti-HA antibody (1:200) diluted in blocking buffer. Eggs were washed in blocking buffer 3× for 10 min, followed by incubation at RT for 30 min with a secondary antibody, Alexa Fluor 488 goat anti-mouse IgG (H+L) (1:400) diluted in blocking buffer. Fluorescence signals were visualized using a laser-scanning confocal microscope (Nikon A1 Resonant Confocal with six-color TIRF) fitted with a 63×, 1.4 NA oil-immersion objective lens.

## Statistical analysis

Comparisons for statistical significance of experimental values between treatments and experiments were performed in three or more experiments performed on different batches of eggs in most studies. Given the number of eggs needed, WB studies were repeated twice. Prism-GraphPad software was used to perform the statistical comparisons that include unpaired Student's *t*-tests, Fisher's exact test, and one-way ANOVA followed by Tukey's multiple comparisons, as applicable, and the production of graphs to display the data. All data are presented as mean ± SD. Differences were considered significant at $p < 0.05$.

## Acknowledgements

We thank Ms. Changli He for technical support and Dr. James Chambers for support with confocal microscopy support. We thank all members of the Fissore lab for useful discussions and suggestions. We thank Jan B Parys, KU Leuven, Belgium, for initial discussions and advice.

## Additional information

### Funding

| Funder | Grant reference number | Author |
|---|---|---|
| Japan Society for the Promotion of Science | | Hiroki Akizawa |
| Eunice Kennedy Shriver National Institute of Child Health and Human Development | RO1 HD092499 | Rafael A Fissore |
| National Institute of Food and Agriculture | 2021-06893 | Rafael A Fissore |

The funders had no role in study design, data collection and interpretation, or the decision to submit the work for publication.

## Author contributions
Hiroki Akizawa, Data curation, Formal analysis, Funding acquisition, Validation, Investigation, Visualization, Methodology, Writing - original draft, Writing - review and editing; Emily M Lopes, Data curation, Formal analysis, Investigation; Rafael A Fissore, Conceptualization, Resources, Supervision, Funding acquisition, Validation, Writing - original draft, Project administration, Writing - review and editing

## Author ORCIDs
Hiroki Akizawa ⬥ http://orcid.org/0000-0003-1091-5629
Rafael A Fissore ⬥ http://orcid.org/0000-0001-5655-0915

## Ethics
The University of Massachusetts Institutional Animal Care and Use Committee (IACUC) approved all animal experiments and protocols (#4650).

Reviewer #1 (Public Review): https://doi.org/10.7554/eLife.88082.3.sa1
Reviewer #2 (Public Review): https://doi.org/10.7554/eLife.88082.3.sa2
Reviewer #3 (Public Review): https://doi.org/10.7554/eLife.88082.3.sa3
Author Response https://doi.org/10.7554/eLife.88082.3.sa4

# Additional files

## Supplementary files
• MDAR checklist

## Data availability
All data generated or analyzed during this study are included in the manuscript and supporting files. Source data files have been provided for Figures 2 and 4.

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
