## [Editor Report · eLife assessment]

This article reports an **important** series of results showing the relationship between oscillatory zinc and calcium fluctuations during egg activation and fertilization. **Compelling** evidence using several complimentary approaches provides further insight into the signals for proper egg activation that underpin successful fertilization and embryo development. The findings are significant because they may lead to improvements in assisted reproduction methods.

---

## [Referee Report · Reviewer #1 (Public Review)]

The study utilizes a variety of methods, chemical and expressed probes, caged release of IP3, as well as oocytes with mutations that alter zinc availability, that provide an elegant examination of how zinc deficiency and zinc excess modulate the transient and cyclic release of calcium during egg activation. In this manuscript, the authors sought to determine if there is any interplay between zinc and calcium, two divalent cations that have been demonstrated to have important roles during fertilization. They employ agents that disrupt normal zinc homeostasis and then monitor the resulting calcium oscillations during egg activation. If zinc was made unavailable via chelation with TPEN, then the calcium oscillations halted. This occurred regardless of the activation method, which included ICSI, PLCζ, Acetylcholine, strontium chloride, and thimerosal. This phenotype could be rescued by introducing zinc back into the egg via an ionophore, such as zinc pyrithione; however, too much zinc pyrithione also halted calcium oscillations. Taken together, these two results demonstrate that there is a threshold level of zinc that is required for proper calcium oscillations to occur.

Furthermore, the authors sought to understand how zinc affects the IP3 receptor, IP3R1. IP3R1 is the receptor that modulates the release of calcium from the endoplasmic reticulum. The authors cited a previous study that identified zinc binding sites on IP3R1. The authors highlight that there exist no studies regarding the regulation of IP3R1 by zinc; however, such studies were cited for a similar calcium channel, the RyRs. The authors use thapsigargin to inhibit the SERCA pump, leading to calcium leak from the IP3R1. TPEN blunted the amount of calcium leaked from the ER following treatment, suggesting that zinc occupancy is necessary for IP3R1 function.

The results of these experiments support the authors conclusions that zinc is essential for the IP3R1-mediated release of calcium in an oscillatory manner during egg activation. These results provide further insight into signals necessary for proper egg activation and the ultimate success of the resulting embryo.

---

## [Referee Report · Reviewer #2 (Public Review)]

The manuscript describes more fully the relationship between zinc fluxes and calcium oscillations during egg activation by directly manipulating the level of zinc ions inside and outside the cell with chelators and ionophores and then measuring resulting changes in Ca++ oscillations. The authors have provided solid evidence consistent with their hypothesis that zinc ions regulate Ca++ oscillations by directly modulating the gating of the IP3-R which is the main calcium channel responsible for calcium release into the cytoplasm. The authors employ well established methods of calcium measurement along with various chelators, ionophores and a wide variety of methods that cause egg activation to demonstrate that an optimal level of zinc ions are required for Ca++ oscillations to occur.

Helpfully, the authors provide a model to explain their observations in Figure 7. In the model, the increase in zinc during maturation is permissive for later IP3-R gating in response to activation. The experiments with TPEN solidly demonstrate that Zn is required because lowering free zinc, (as indicated by Fluozin staining), abrogates Ca++ oscillations. This is true regardless of the method of activation. What is not clearly described in the model or in the manuscript is whether the levels of zinc at MII are simply permissive for IP3-R gating or whether there is a more acute relationship between zinc fluxes and Ca++ oscillations. In the original paper describing the zinc spark (Kim et al., ACS Chem Biol 6:716-723), the authors show that zinc efflux very closely mirrors Ca++ oscillations suggesting a more active relationship.

The role of TRPv3 and Trpm7 in Zn homeostasis during egg activation seems to be minor. Labile zinc accumulation, as measured by fluozin-3 staining, is reduced in the dKO eggs, but is this modest decrease in labile Zn responsible for the changes in Ca release after Tg treatment? There is an increase in the amplitude of Tg-induced Ca release in dKO eggs. This argues that there is not an inhibitory effect in the dKO mice. That it takes a little longer to reach Ca peak could be due to the greater amount of Ca being released.

The effect of PyT on the apparent rise in cytoplasmic Ca++ in Figure 6 is interpreted as caused by an artifact of high Zn concentrations. However, it is not clear that 0.05 uM PyT would necessarily increase cytoplasmic Zn to the point where FURA-2 fluorescence would increase. A simpler interpretation is that PyT allows sufficient Zn to enter the cell and keeps the IP3-R channels open causing a sustained rise in cytoplasmic Ca and preventing oscillations in Ca++. This interpretation would also preclude inhibitory effects of high Zn concentrations as shown in Figure 7 which may or may not be present but are simply speculation.

Overall, this study significantly advances our understanding of egg activation and could lead to better ways of in vitro egg activation in women undergoing assisted reproduction.

---

## [Referee Report · Reviewer #3 (Public Review)]

This study investigated the role of Zn2+ on the maintenance of Ca2+ oscillation upon fertilization. TPEN was used to reduce the level of available Zn2+ in fertilized oocytes and different inhibitors were used to pinpoint the mechanistic involvement of intracellular Zn2+ on the maintenance of Ca2+ oscillation. As also stated in the manuscript, previous studies have demonstrated the role of Zn2+ for the successful completion of meiosis/fertilization. The manuscript expands our understanding of fertilization process by describing how the level of Zn2+ regulates Ca2+ channels and stores. The manuscript is well-organized and the topic is important in early embryo development fields.

The authors added more information to the manuscript based on reviewers' comments. The quality of the manuscript has been improved and the study addresses important questions in mammalian fertilization.

---

## [Author Response]

The following is the authors’ response to the original reviews.

**Reviewer #1:**

We thank the reviewer for the positive evaluation of our manuscript. We have closely examined the issues raised, and below we offer a point-by-point response to each comment. In the revised manuscript below, all the introduced changes are marked with red font.

1. There may be a general typo concerning micromolar and millimolar…

Response 1: The reviewer is correct, and during the reformatting of the manuscript, in some portions of the manuscript, the units used to indicate TPEN concentrations, always µM, were switched to mM. We have corrected those mistakes.

1. In Figure 1C/Lines 150-152, the authors use DTPA and EDTA as extracellular chelators for zinc… Was the amount of zinc in the media measured and determined to be below the amount of chelator used? Additionally, these chelators are not specific for zinc, but can bind other divalent cations including calcium. Even though zinc binds more tightly than calcium to these chelators, by mass action calcium and magnesium ions may outcompete DTPA and EDTA, leaving zinc availability unperturbed. How do the authors take these interactions into account to determine that chelation of extracellular zinc has no effect on intracellular calcium oscillations? The best way to test this is to use zinc responsive fluorescent probes in a sample of the calcium- and magnesium-replete medium and see if the addition of the DTPA or EDTA alters zinc fluorescence in the cuvette.

Response 2: We tested several conditions to determine the effect of chelators on the zinc concentration of the monitoring media using commercially available Zn2+ probes. The fluorescent zinc probe FluoZin3 added extracellularly shows high fluorescence, consistent with trace amounts of zinc and possibly non-specific bindings of other cations.

Further, the media tested was replete with the concentrations of Ca2+ and Mg2+ in TLHEPES. To establish if the non-permeable external chelators we used could bind external Zn2+ despite the high concentrations of Ca2+ and Mg2+, we followed the reviewer’s suggestion of adding the chelators to the complete media in the presence of FluoZin3. The addition of EDTA caused a protracted, ~5 min, but significant decrease in FluoZin3’s fluorescence, suggesting it is effective at removing external Zn2+ despite the presence of other divalent cations (Author response image 1A). We used a second approach where we added the chelator in the presence of nominal concentrations of Ca2+ and Mg2+ to increase the chelators’ chances to find and chelate Zn2+ (Author response image 1B). Then, we injected mPlcζ mRNA, which initiated persistent but low-frequency oscillations, as expected due to the lack of external Ca2+. Remarkably, upon restoring it, the responses became of high frequency, and upon increasing Mg2+, they acquired the regular pattern, consistent with Mg2+’s inhibition of channels that mediate Ca2+ influx. These results show that the chelation of extracellular zinc does not replicate TPEN’s effect, which suggests that TPEN’s abrupt and inhibiting ability on Ca2+ oscillations is most likely due to the 43 chelation of internal Zn2+.

**Author response image 1. sa4fig1:** Cell-impermeable chelators effectively reduce Zn2+ levels in external media but do prevent initiation or continuation of Ca2+ oscillations. (A) A representative trace of FluoZin3 fluorescence in replete monitoring media (TL-HEPES). The media was supplemented with cell-impermeable FluoZin-3, and after initiation of monitoring, the addition of EDTA (100 μM) occurred at the designated point (triangle). (B) The left black trace represents Ca2+ oscillations initiation by injection of mPlcζ mRNA (0.01 μg/μl). The oscillations were monitored in Ca2+ and Mg2+-free media and in the presence of EDTA (110 μM) to chelate residual divalent cations derived from the water source or reagents used to make the media. The right red trace represents the initiation of oscillations as above, but after a period indicated by the black and green bars, Ca2+ and Mg2+ were sequentially added back.

Noteworthy, low EDTA concentrations, 10-µM, have been used to enhance in vitro culture conditions of mammalian embryos. In fact, it is the key ingredient to overcome the two-cell block that initially prevented the in vitro development of zygotes srom inbred strains. It is unknown how EDTA mediates this effect, which is detectable in Ca2+ and Mg2+ replete media and is only effective when placed extracellularly, but it has been attributed to its ability to chelate toxic metals introduced as impurities by other media components; one study demonstrated that the Zn2+ present in the oil used to overlay the culture medium micro drops was the target (Erbach et al., Human Reproduction, 1995, 10, 3248-54). We included some of these points in the revised version of the manuscript and added this figure as Supplementary Figure 1.

1. The reviewer noted that while dKO eggs showed reduced labile zinc levels, the amount of total zinc is not determined. Further, the response to thapsigargin in dKO eggs didn’t phenocopy the profile in eggs treated with TPEN. The reviewer argued that without further experimentation, such as comparing polar body extrusion and egg activation rate between WT and dKO, it seems to be a stretch to state that these eggs are zinc deficient.Response 3: We agree that the statement, ‘zinc deficient,’ is an overstatement without determining the total zinc levels and associated phenotypes. Therefore, in the revised version of the manuscript, we referred to dKO-derived eggs and embryos as “low-level labile Zn2+ eggs”. Our follow-up studies show that eggs from dKO females seem to undergo egg activation events, such as the timing and rate of second polar body extrusion and pronuclear formation, with a similar dynamic to WT females. Hence, we estimate that the labile Zn2+ levels in dKO eggs are not as low as those of WT eggs treated with TPEN. Consequently, these intermediate zinc levels may have subtle effects, such as changing the Thapsigargin-induced Ca2+ release through the IP3R1 without causing widespread inhibition of cellular events observed after TPEN. We would argue that this approach is significant because it can distinguish how the different cellular events and proteins and enzymes have distinct affinities or zinc requirements and, in this case, start uncovering the channel(s) present in oocytes and eggs that may contribute to regulating zinc homeostasis.1. The reviewer pointed out that since zinc is not redox active, it is unclear how zinc could be modifying cysteine residues of IP3R1.The reviewer suggested the possibility that excess zinc is binding to the cysteines and preventing their oxidation leading to the inhibition of the IP3R1 by blocking the channel, thereby preventing calcium release.

Response 4: The reviewer correctly points out that the mechanism(s) whereby excess Zn2+ modifies the IP3R1 function is undetermined in our study. Further, our description of ‘modifying’ is ambiguous and could be misinterpreted. Data in the literature, some of which we cite in the manuscript, shows that “oxidation of cysteine residues enhances receptor’s sensitivity to ligands in various cell types”. Zn2+ preferentially binds to reduced cysteine residues, and thus, we agree with the proposed reviewer's suggestion that “excess zinc may occupy reduced cysteine residues, preventing their oxidization required to sensitize the receptor”. As noted by the reviewer, we cannot rule out that it might be directly blocking the IP3R1 channel. We have modified the corresponding paragraphs in the Discussion.

1. Line 80 and 411, there are three other reports demonstrate the zinc reallocation to the egg shell or ejection as the zinc spark; Zebrafish: Converse et al. in Sci. Reports 10, 15673 (2020); X. lavis: Seeler et al. in Nature Chem. 13, 683-691 (2021), *C. elegans*: Mendoza et al. in Biology of Reproduction 107(2):406-418 (2022).

Response 5: Thank you for pointing this out, and we have added these references.

1. Line 129, when discussing that Zn2+ concentrations are reduced after TPEN as visualized by FluoZin-3, the authors should cite the article in which FluoZin-3 was first reported and this result was demonstrated initially: "Detection and Imaging of Zinc Secretion from Pancreatic β-Cells Using a New Fluorescent Zinc Indicator" by Gee et al. J. Am. Chem. Soc 124, 5, 776-778.

Response 6: Thank you for pointing this out, and we have added this reference.

1. In Figure 1E/Table 1 the authors evaluated if TPEN supplementation affects meiosis and pronuclear formation; however, the timing of TPEN treatment is unclear. When was TPEN introduced? Were the eggs left in the same media containing TPEN following fertilization, or were they transferred to different media?

Response 7: Thank you for pointing this out, and we have noted the time of the addition in the figure and text.

1. Line 1011 and 1012, ZnTP should be ZnPT.

Response 8: Thank you for pointing this out, which is now corrected.

**Reviewer #2:**
1. The reviewer raises the question of whether a more complex relationship could exist between the levels of zinc in MII eggs by indicating, “a more active relationship such that zinc efflux associated with each calcium spike could be necessary for terminating the Ca spike by depleting cytoplasmic zinc.” The reviewer also states, “Perhaps, rather than simply a permissive role, the normal Zn fluxes during activation may be acutely changingIP3-R gating sensitivity.”

Response 1: We agree that the demonstration that TPEN dose-dependently delays and consistently terminates ongoing Ca2+ rises perhaps reflects a more nuanced relationship between cytoplasmic labile zinc concentrations, Ca2+ oscillations, and IP3R1 function. Uncovering the precise nature of this relationship would require additional studies, such as determining the impact of TPEN on IP3 binding to its cognate receptor, regulation of channel gating, and more in-depth functional-structural experiments. However, these studies will demand time and complex experimental design and are beyond the scope of the current work. Nevertheless, they are excellent suggestions for future studies.

We would argue against the reviewer’s suggestion that “zinc sparks directly contribute to shaping the oscillations.” Zn2+ released during the sparks is not labile, but Zn2+ bound to cortical granules-resident proteins, most of which are inaccessible to the cytosol and hence to IP3R1s and should not perturb its function. We examined (data not shown) that the levels of cytosolic labile Zn2+, as assessed with FluoZin3, remained steady for over three hours of Plcζ mRNA-initiated oscillations. Further, because the Zn2+ sparks cease after the third or fourth Ca2+ rise, it would mean, at the very least, that this mechanism only operates on the first few responses. Thus, while the change of cytosolic Ca2+ concentrations triggers the Zn2+ sparks, we argue that the opposite influence is unlikely to hold true.

1. The reviewer also pointed out that the role of Trpv3 and Trpm7 in Zn2+ homeostasis seems to be minor and that the effects of genetic deletion of those channels are not as clear as those obtained by TPEN. Given that dKO eggs make it to the MII and release more but not less calcium upon thapsigargin than control despite the lowered labile Zn2+ level, the reviewer speculated that the loss of those channels changes calcium gating independent of Zn2+ concentration.

Response 2: TRPV3, TRPM7, and Cav3.2 are the three channels identified to permeate Ca2+ during oocyte maturation and egg activation in mice. We and other groups have observed that in oocytes and eggs, these channels partly compensate for the absence of each other because the deletion of these channels individually has a limited effect on Ca2+ oscillations and fertility. Thus, in the case of oocytes from Trpv3 and Trpm7 dKO animals, the other plasma membrane channel(s), most likely Cav3.2, is plausibly compensating, and its enhanced function underlies the increased Ca2+ response to Thapsigargin.

Nevertheless, the slower time to the peak and the lesser steep rise of the Thapsigargin induced rise suggest a negative impact of the dKO environment on IP3R1’s ability to mediate Ca2+ release. Based on the rest of the results in the manuscript, we attribute this change to the lower levels of labile Zn2+ in dKO eggs.

1. Lastly, the reviewer noted the upregulation of the Fura-2AM following addition of ZnPT. The reviewer indicated that 0.05 uM ZnPT might not increase intracellular Zn2+ to change Fura-2 fluorescence, but it might be sufficient Zn2+ to enter the cell and keep the IP3R1 channels open causing a sustained rise in cytoplasmic calcium and preventing oscillations. Further, if this interpretation holds true, the inhibitory effects of high Zn2+ on IP3R1’s gating shown in figure 7 would be precluded.

Response 3: We acknowledge that the increased levels of Fura-2 fluorescence following the addition of ZnPT could be due to the increased Zn2+ levels acting on IP3R1, increasing its open probability, and elevating cytosolic Ca2+ levels. We have added this consideration to the discussion. Nevertheless, our evidence suggests that this is unlikely because, as shown in Figure 6 H, I, the ER-Ca2+ levels as assessed by D1ER recordings did not change following the addition of ZnPT, whereas Rhod-2 fluorescence did, suggesting that the two events are seemingly uncoupled. Further, constant leak from the ER and extended high cytosolic Ca2+ would lead to egg activation or cell death, neither of which changes were observed.

**Reviewer #3:**

The reviewer noted that the present study deepened the understanding of the role of zinc in regulating calcium channels and stores at fertilization beyond the previously known Zn2+ requirement in oocyte maturation and the cell cycle progression. We appreciate these comments.

1. Fig. 1. The reviewer wondered why we selected 10 μM TPEN for most of the experiments in the manuscript. The reviewer noted this concentration only stopped the Ca2+oscillations in just half of the eggs after ICSI.

Response 1: We used 10-μM TPEN throughout the study because it blocked ~50% of the oscillations of a robust trigger of Ca2+ responses such as ICSI and reduced the frequency in the remaining eggs. This concentration of TPEN abrogates and prevents the responses by milder stimuli, such as Acetylcholine and SrCl2. Importantly, thimerosal and Plcζ mRNA overcome the inhibition by 10μM but not 50-μM TPEN. However, 50μM TPEN inactivates Emi2, a Zn2+-dependent enzyme, causing parthenogenic activation and cell cycle progression, and we wanted to avoid this confounding factor. Therefore, we determined 10-μM is a “threshold” concentration and selected it for the remaining studies. We also reasoned that it would allow the detection of more subtle effects of reducing the levels of labile zinc, causing a milder inhibition of IP3R1 sensitivity and a progressive delay or modification of the responses to other agonists rather than fully abrogating them, which is the case with higher concentrations.

1. Line131 - no concentration of TPEN stated? Or 'the addition of different concentrations of TPEN"?

Response 2: We have corrected this. We have now added 50-100 µM concentrations.

1. Line 146 - instead of TPEN, all TPEN concentrations?

Response 3: We have added these corrections, as at the concentrations we tested here, 5μM TPEN and above, all caused a reduction in the baseline of Fura-2 fluorescence.

1. Line 1046 - 'We submit'? Propose?

Response 4: We have replaced the word submit for propose. Thank you for the suggestion.